# Prediction of base editor off-targets by deep learning

Chengdong Zhang[1,2,3,9], Yuan Yang [1,2,9], Tao Qi[1], Yuening Zhang[4], Linghui Hou[1], Jingjing Wei[1], Jingcheng Yang[1], Leming Shi [1], Sang-Ging Ong [5,6], Hongyan Wang[1,9], Hui Wang[2] ✉, Bo Yu [1] ✉ & Yongming Wang [1,7,8] ✉

Due to the tolerance of mismatches between gRNA and targeting sequence, base editors frequently induce unwanted Cas9-dependent off-target mutations. Here, to develop models to predict such off-targets, we design gRNA-off-target pairs for adenine base editors (ABEs) and cytosine base editors (CBEs) and stably integrate them into the human cells. After five days of editing, we obtain valid efficiency datasets of 54,663 and 55,727 off-targets for ABEs and CBEs, respectively. We use the datasets to train deep learning models, resulting in ABEdeepoff and CBEdeepoff, which can predict off-target sites. We use these tools to predict off-targets for a panel of endogenous loci and achieve Spearman correlation values varying from 0.710 to 0.859. Finally, we develop an integrated tool that is freely accessible via an online web server http://www.deephf.com/#/bedeep/bedeepoff. These tools could facilitate minimizing the off-target effects of base editing.

Base editors enable the programmable conversion of a single nucleotide (nt) in the mammalian genome and have a broad range of research and medical applications. They are fusion proteins that include a catalytically impaired Cas9 nuclease (Cas9[D10A]) and a nucleobase deaminase[1,2]. Cas9[D10A] nuclease and a ~100 nt single-guide RNA (sgRNA) form a Cas9[D10A]-sgRNA complex, recognizing a 20 nt target sequence followed by a downstream protospacer adjacent motif (PAM)[3–5]. Once the Cas9[D10A]-gRNA complex binds to target DNA, it opens a single-stranded DNA loop[3]. The nucleobase deaminase modifies the single-stranded DNA within a small ~5nt window at the 5′ ends of the target sequence[1,2]. Two classes of base editors have been developed: cytidine base editors (CBEs) convert target C:G base pairs to T:A[1], and adenine base editors (ABEs) convert A:T to G:C[2]. Base

editors have been successfully used in diverse organisms, including prokaryotes, plants, fish, frogs, mammals, and human embryos[6–11].

Although there exist several tools for base editor on-target efficiency prediction[12–14], a major safety concern is that base editors can induce unwanted off-target effects, including Cas9-independent off-target and Cas9-dependent off-target effects. Cas9-independent off-target effects are caused by random deamination, which can be minimized by using modified deaminases[15,16], or a cleavable deoxycytidine deaminase inhibitor[17]. Cas9-dependent off-target effects are caused by tolerance to mismatches between the gRNA and targeting sequence[18,19]. Cas9-dependent off-targets can be minimized by careful gRNA selection, but experimental evaluation of off-targets is time-consuming[18–20], prompting us to develop in silico tools for off-target

---

[1]Center for Medical Research and Innovation, Shanghai Pudong Hospital, Fudan University Pudong Medical Center; State Key Laboratory of Genetic Engineering, School of Life Sciences, Zhongshan Hospital, Fudan University, Shanghai 200438, China. [2]State Key Laboratory of Oncogenes and Related Genes, Center for Single-Cell Omics, School of Public Health, Shanghai Jiao Tong University School of Medicine, 200025 Shanghai, China. [3]Department of Clinical Oncology, Taihe Hospital, Hubei University of Medicine, Shiyan, China. [4]SJTU-Yale Joint Center for Biostatistics and Data Science, (Department of Bioinformatics and Biostatistics, School of Life Sciences and Biotechnology) Shanghai Jiao Tong University, Shanghai 200240, China. [5]Department of Pharmacology and Regenerative Medicine, University of Illinois College of Medicine, Illinois, USA. [6]Division of Cardiology, Department of Medicine, University of Illinois College of Medicine, Illinois, USA. [7]Department of Cardiology, The First Affiliated Hospital of Zhengzhou University, Zhengzhou, Henan 450052, China. [8]Shanghai Engineering Research Center of Industrial Microorganisms, Shanghai, China. [9]These authors contributed equally: Chengdong Zhang, Yuan Yang. ✉e-mail: huiwang@sibs.ac.cn; paul.yubo@gmail.com; ymw@fudan.edu.cn

prediction. We previously used a high-throughput strategy for gRNA-target library screening for SpCas9 activity[21].

In this study, we design libraries of gRNA-off-target pairs and perform a high-throughput screen, obtaining valid 54,663 and 55,727 editing efficiencies for ABE and CBE, respectively. The resulting datasets are used to train deep learning models, resulting in ABEdeepoff and CBEdeepoff, which can predict editing efficiency at potential off-targets.

## Results

### A guide RNA–target pair strategy for testing of editing efficiency at off-target sites

To investigate mutation tolerance at off-target sites, we designed two gRNA-target pair libraries (Fig. 1a, Supplementary Fig. 1). These gRNAs were randomly selected from our previously designed gRNA library targeting the human genome[21]. The ABE library contains 91,287 gRNA-target pairs distributed among 1383 gRNA groups (Supplementary Data 1). The CBE library contains 91,174 gRNA-target pairs distributed among 1378 gRNA groups (Supplementary Data 2). Each group contains one gRNA-on-target pair and multiple gRNA-off-target pairs. The mutation type included mismatches (1-6 bp mismatches per off-target, 71,099 pairs for ABE, and 70,594 pairs for CBE), deletions (1-2 bp per target, 6521 pairs for ABE, and 6759 pairs for CBE), insertions (1-2 bp per target, 11,396 pairs for ABE, and 11,562 pairs for CBE) and mismatches mixed with insertions/deletions (indels, 1-2 bp mismatches plus 1-2 bp indels, total mutations are 2-3 nt; 888 pairs for ABE, and 881 pairs for CBE). The GC content of the ABE library accounted for 52.99%; the GC content of the CBE library accounted for 55.15%. The GC content of the ABE library positionally varied from 38.83 at position 7 to 65.80% at position 20; the GC content of the CBE library positionally varied from 43.40% at position 14 to 66.04% at position 20 (Supplementary Data 3 and 4).

Next, we generated a panel of single-cell-derived clones that stably express ABE or CBE base editors. Optimized versions of base editors (ABEmax for ABE; AncBE4max for CBE)[22] were stably integrated into the genome using the Sleeping Beauty (SB) transposon system (Supplementary Fig. 2a)[23–25], and single-cell clones were formed. We tested the conversion efficiency in these clones and selected an efficient clone for each base editor (Supplementary Fig. 2b, c).

We packaged the gRNA-target pair library into lentiviruses and transduced them into base editor-expressing cells. Five days after transduction, genomic DNA was extracted, and synthesized off-targets were PCR-amplified for deep sequencing. Deep sequencing results revealed that A to G conversion for ABE and C to T conversion for CBE occurred (Fig. 1b). In this study, editing efficiency was defined as the number of edited reads divided by the number of total reads. Only reads number over 100 was considered valid data. The screening assay was experimentally repeated twice, and editing efficiency in two independent replicates showed a high correlation (Pearson correlation, 0.970 for ABE and 0.994 for CBE, Supplementary Fig. 3a–d). This paves the way for merging the two replicates to expand the training dataset. An off-target efficiency was calculated as the average of two replicates (Supplementary Fig. 3e, f). For example, if an off-target efficiency was 0.5 in replicate 1 and 0.7 in replicate 2, the final off-target efficiency was 0.6.

We obtained ABE off-target efficiencies of 54,663 varied from 0% to 100%; we obtained ABE on-target efficiencies of 1110 varied from 13.7% to 97.6% (Supplementary Data 1). Similarly, we obtained CBE off-target efficiencies of 55,727 varied from 0% to 100%; we obtained CBE on-target efficiencies of 1076 varied from 28.9% to 100% (Supplementary Data 2). Since the on-target editing efficiencies vary over a wide range, we used the ratio of off-target efficiency to on-target efficiency (off:on-target ratio) to normalize the off-target efficiency.

The large-scale datasets generated here allowed us to analyze the effects of mutation type on the off-on-target ratio. For both base editors, all mutation types have a negative impact on the off:on-target ratio on average (off:on-target ratio of 0.673 for ABE and 0.695 for CBE). The off:on-target ratio decreased with an increasing number of mismatches (Fig. 1c, d). Deletions had a stronger influence than insertions and mismatches. The overall off:on-target ratio of ABE was ranked as 1mis > 1ins > 1del > 2mis > 2ins > mix > 3mis > 2del > 4mis > 5mis > 6mis (Fig. 1c); the overall off:on-target ratio of CBE was ranked as 1mis > 1ins > 2mis > 1del > 2ins > mix > 3mis > 2del > 4mis > 5mis > 6mis (Fig. 1d).

Next, we investigated the positional effects of mutation on the off:on-target ratio. For both base editors, on average, mutations at target positions 1-10 had higher tolerance compared to mutations at target positions 11–20. Due to the average off:on-target ratio in 1mis being 0.844 for ABE and 0.888 for CBE, we considered that an off:on-target ratio ≤ 0.8 represented a significant decrease in off-target editing efficiency compared to its perfectly matched targets. In our research, one mismatch significantly decreased the off:on-target ratio at positions 14−16; one insertion significantly decreased the off:on-target ratio at positions 11−18; one deletion significantly decreased the off:on-target ratio at positions 3−7 and 10−20 (Fig. 1e, f). We observed that one mismatch and one insertion at positions 19−20 did not influence the off-on-target ratio (Fig. 1e, f). Previous studies have shown that some single mismatches at position 20 did not significantly influence the off-target editing efficiency[19,20]. We further used an independent T-test to compare the off:on-target ratio across different positions, and a Bonferroni correction for multiple tests was applied. It is obvious that the statistical results (p-value) indicate significant differences in most positions within the 1mis, 1ins, and 1del mutation types (Supplementary Data 5 and 6).

Next, we investigated the positional effects of every single-nucleotide mutation on the off:on-target ratio. The off:on-target ratio was standardized by the Z-score, where values higher than 1 or lower than −1 signified that the mutation had an important contribution to the off:on-target ratio than the average. The results revealed that ABE and CBE had similar z-score distribution (Supplementary Fig. 4a, b). For a given mutation type, all four nucleotides showed similar z-score distribution. For example, z-scores for one nucleotide deletion at positions 1–10 were over zero, demonstrating that one nucleotide deletion at these positions had a less negative impact on the editing efficiency than average (i.e., more tolerant to mutations than average). Z-scores for one nucleotide deletion at positions 11–20 were lower than zero, demonstrating that one nucleotide deletion at these positions had more negative impact on the editing efficiency than average (Supplementary Fig. 4a, b). We further investigated the influence of two mismatches on the off:on-target ratio. Two mismatches had a strong influence on the off:on-target ratio when they both occurred in the seed region (1 - 9 nucleotides proximal to the PAM) (Supplementary Fig. 5a, b). Two mismatches on the seed region have been shown to strongly influence the indel efficiency of SaCas9[26] and SpCas9[27].

### Developing models for predicting off:on-target ratio at off-targets

Next, we used the ABE off-target editing datasets to train a fusion embedding-based deep learning model where a gRNA-target pair was considered a unique sequence and used as input (Fig. 2a). The embedding design encoded the input gRNA and off-target sequences in the same scheme and then embedded the representation vectors in the real value matrix spaces with the same weight initialization settings. Both generalization ability and training speed will benefit from sharing the same weight initialization parameter. Besides, an attention mechanism was also applied in the model to get more representative features from the LSTM output to be fed into the fully connected neural network.

We grouped the datasets according to on-target sequences and obtained 1110 ABE gRNA groups. These groups were randomly divided into training and testing datasets by 10-fold "GroupKFold", with 90% (999) of the gRNA groups assigned to the training set and 10% (111) to

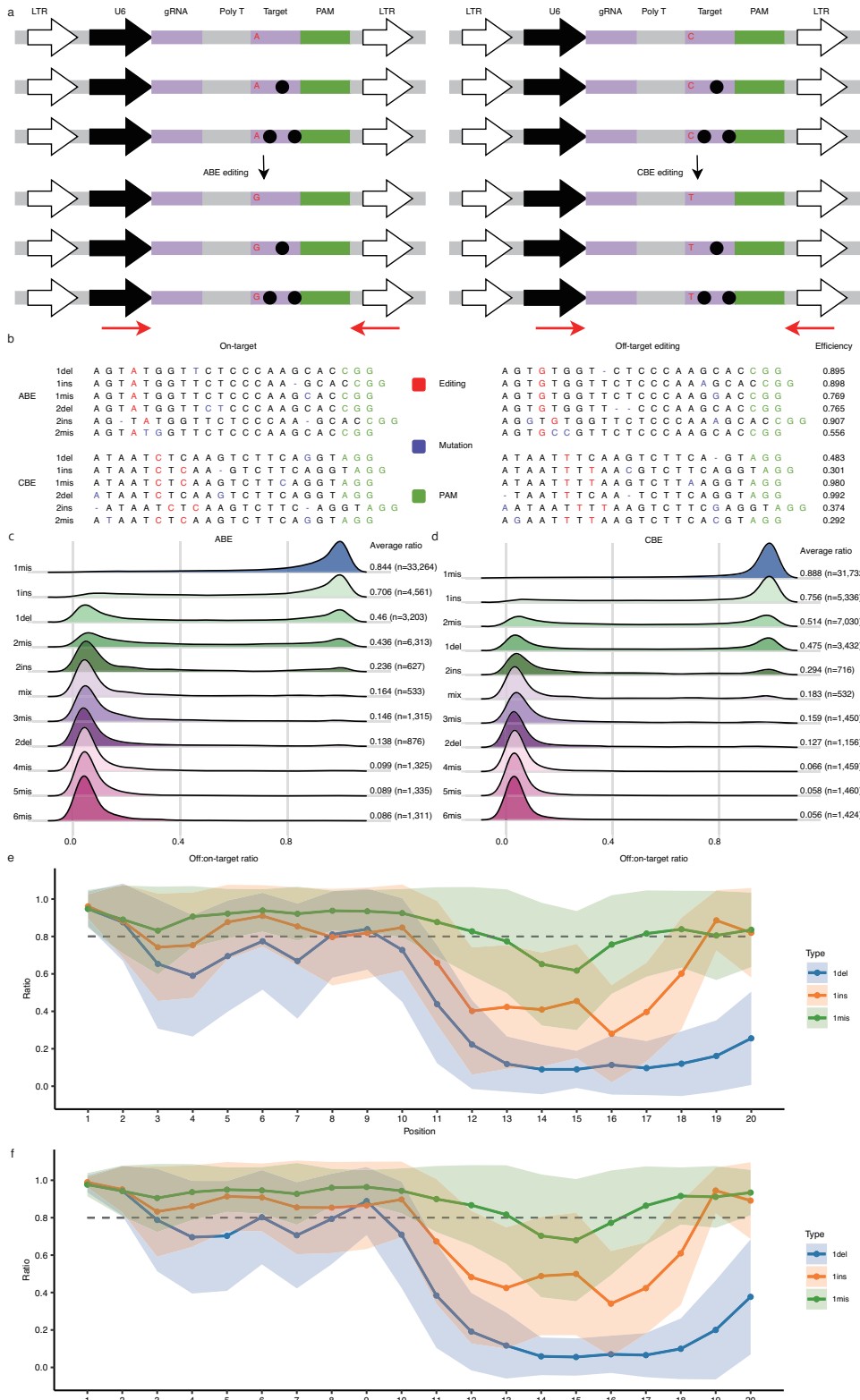

**Fig. 1 | High-throughput screen of off-target efficiencies. a** Schematic of the pooled pairwise library screen for ABE and CBE off-targets. Black dots indicate mutations; Nucleotide conversions are indicated by red letters; red arrows indicate primers for target site amplification. **b** Examples of gRNA-target pair design and their valid editing pattern. Editing efficiencies are shown on the right. Mutations are indicated by purple letters; PAM is indicated by green letters. On: on-target sequence; 1mis/2mis: 1 or 2 bp mismatch; 1del/2del: 1 or 2 bp deletion; 1ins/2ins: 1 or 2 bp insertion. **c, d** Influence of mutation types on the off:on-target ratios. Mutation types are labeled on the left; the average off:on-target ratio (off-target number) is shown on the right. Mix: mismatches mixed with indels. **e, f** Positional effect on off:on-target ratio for 1 bp mismatch ($n$ = 33264 and 31732), 1 bp insertion ($n$ = 4561 and 5336), and 1 bp deletion ($n$ = 3203 and 3432). Data are presented as mean ± s.d. Source data are provided as a Source Data file.

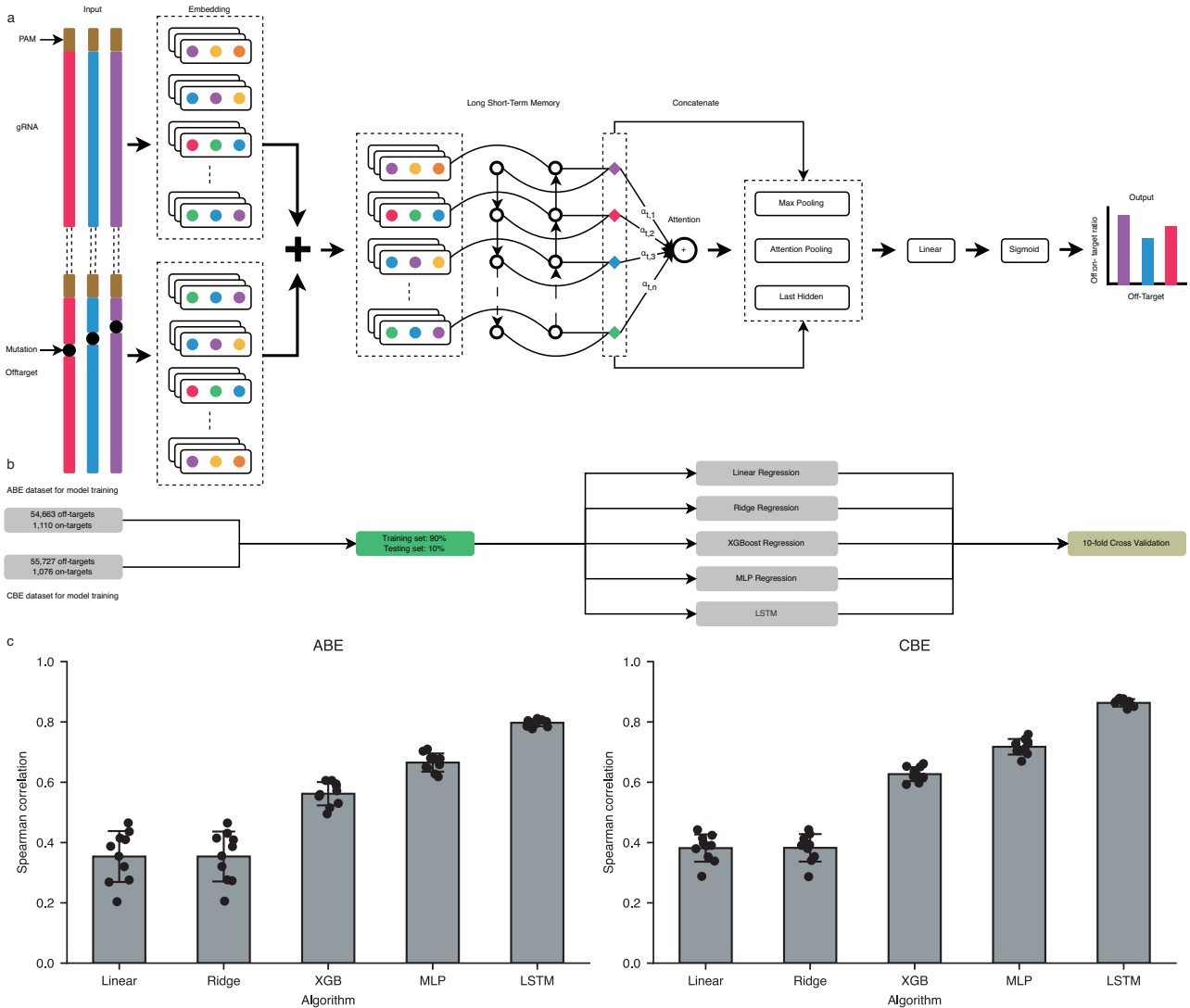

**Fig. 2 | A fusion embedding-based deep learning model for base editor off:on-target ratio prediction. a** Schematic of datasets used for training, validation, and testing schema of the prediction models for ABE and CBE. **b** The gRNA and off-target sequence are paired together as input and embedded in the matrix space to obtain the dense real-valued representation. Their representations are further processed by a matrix summation to obtain the combined features. The combined features are further processed by a two-layer BiLSTM to obtain the hidden representation, which is then processed by the attention mechanism to obtain the Attention Pooling feature. The Attention Pooling was concatenated with the Max Pooling and Last Hidden features. These features serve as the input of the fully connected layers. Finally, a sigmoid transformation is performed to obtain the predicted off:on-target ratios. **c** Performance of different algorithms for the off:on-target ratio prediction revealed by Spearman correlation for ABE and CBE. LSTM is the backbone structure used in our deep learning models. The bar plot shows the mean ± s.d for the Spearman correlation coefficient between predicted and ground truth off:on-target ratios ($n = 10$ testing datasets). Source data are provided as a Source Data file.

the testing set. The internal validation set were randomly sampled from the testing excluded dataset to tune hyperparameters. This resulted in 50,196 training gRNA-off-target pairs and 5577 testing gRNA-off-target pairs (Fig. 2b, Supplementary Fig. 6). The internal testing datasets were never used during the training process. The resulting model was named "ABEdeepoff", which can predict off:on-target ratios at potential off-targets. The model achieved a Spearman correlation (R) of 0.797 ± 0.012 for the testing dataset. We also used the above training datasets to train conventional hand-crafted feature algorithms, including Linear Regression, Ridge Regression, Multiple Perceptron, and XGBoost. The Spearman correlations of these models were lower than that of the ABEdeepoff model (Fig. 2c, Supplementary Data 7). These results demonstrated that the deep learning models outperformed the four conventional algorithms in the ABE dataset.

Next, we used ABEdeepoff to analyze different mutation types in the testing datasets. The model achieved very strong correlations for

1 bp deletion (R = 0.836 ± 0.014); strong correlations for 2 bp mismatch (R = 0.767 ± 0.031), and 1 bp insertion (R = 0.727 ± 0.050); moderate correlations for 1 bp mismatch (R = 0.592 ± 0.026) and 2 bp insertion (R = 0.478 ± 0.195); weak correlations for 2 bp deletion (R = 0.347 ± 0.141), 3 bp mismatches (R = 0.334 ± 0.099) and mixed mutations (R = 0.301 ± 0.246), and very weak correlations for the remaining mutations (Fig. 3a). These results suggested that ABE-deepoff performed well for off-targets with 1–2 bp mismatches, 1–2 bp insertions, and 1 bp deletion. Next, we evaluated the performance of ABEdeepoff with six groups of off-target datasets collected from the literature[18,20], where a series of 1–4 nucleotides of mismatched gRNAs were designed to target endogenous loci, and achieved Spearman correlation values that varied from 0.745 to 0.839 (Fig. 3b, Supplementary Data 8).

In parallel, we used the CBE off-target editing datasets to train the embedding-based deep learning model (Fig. 2a). The 56,803 gRNA-

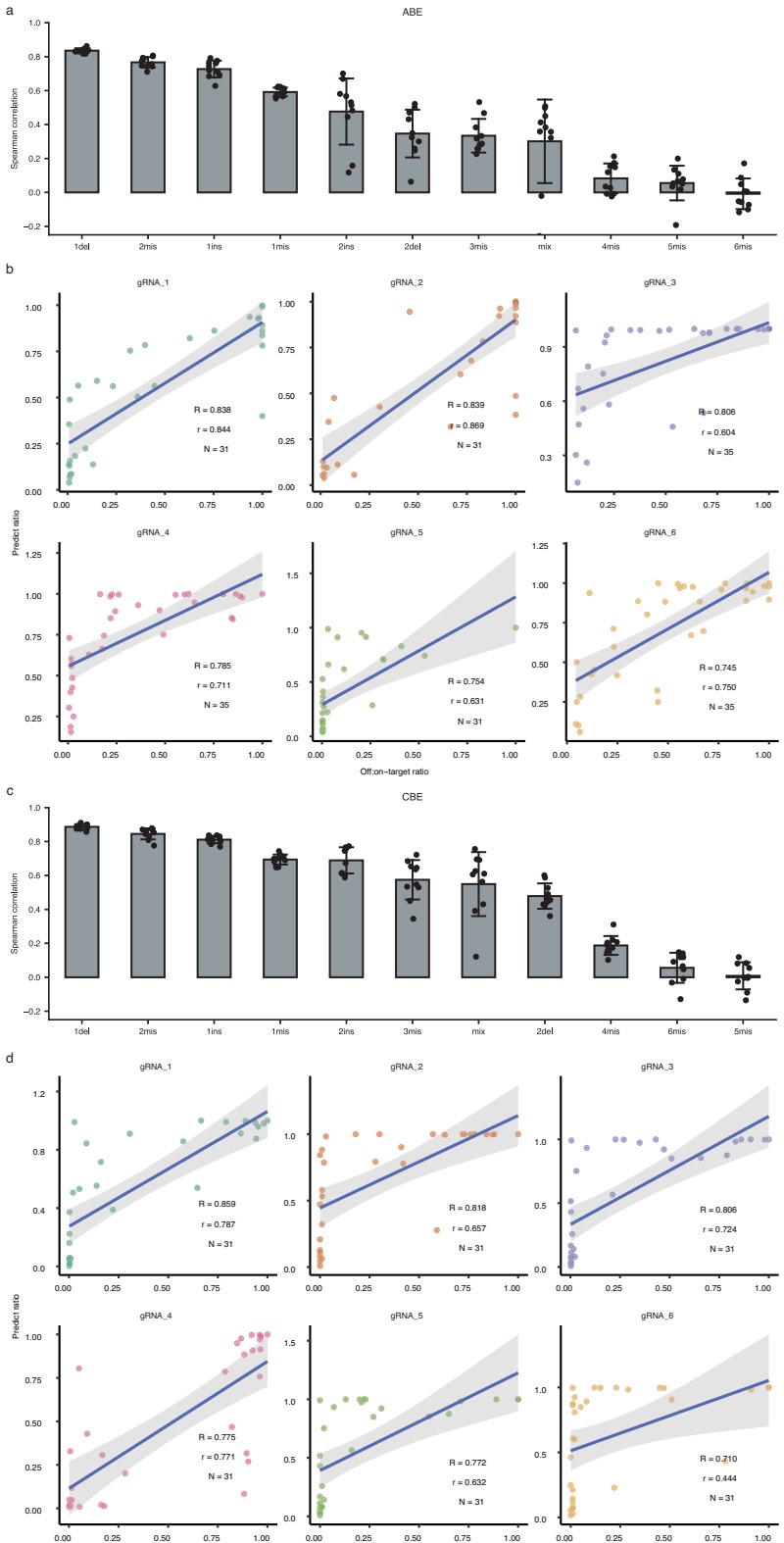

**Fig. 3 | Evaluation of ABEdeepoff/CBEdeepoff for off:on ratio prediction at off-targets. a** Evaluation of ABEdeepoff prediction for different mutation types with a testing dataset ($n$ = 10 testing datasets). Data are presented as mean ± s.d. **b** Evaluation of ABEdeepoff prediction for off:on-target ratio at six groups of external off-targets (N is the number of gRNA and off-target sequence pairs). Data are presented as mean predicted ratio. **c** Evaluation of CBEdeepoff prediction for different mutation types with a testing dataset ($n$ = 10 testing datasets). Data are presented as mean ± s.d. **d** Evaluation of CBEdeepoff prediction for off:on-target ratio at six groups of external off-targets (N is the number of gRNA and off-target sequence pairs). Data are presented as mean predicted ratio. Source data are provided as a Source Data file.

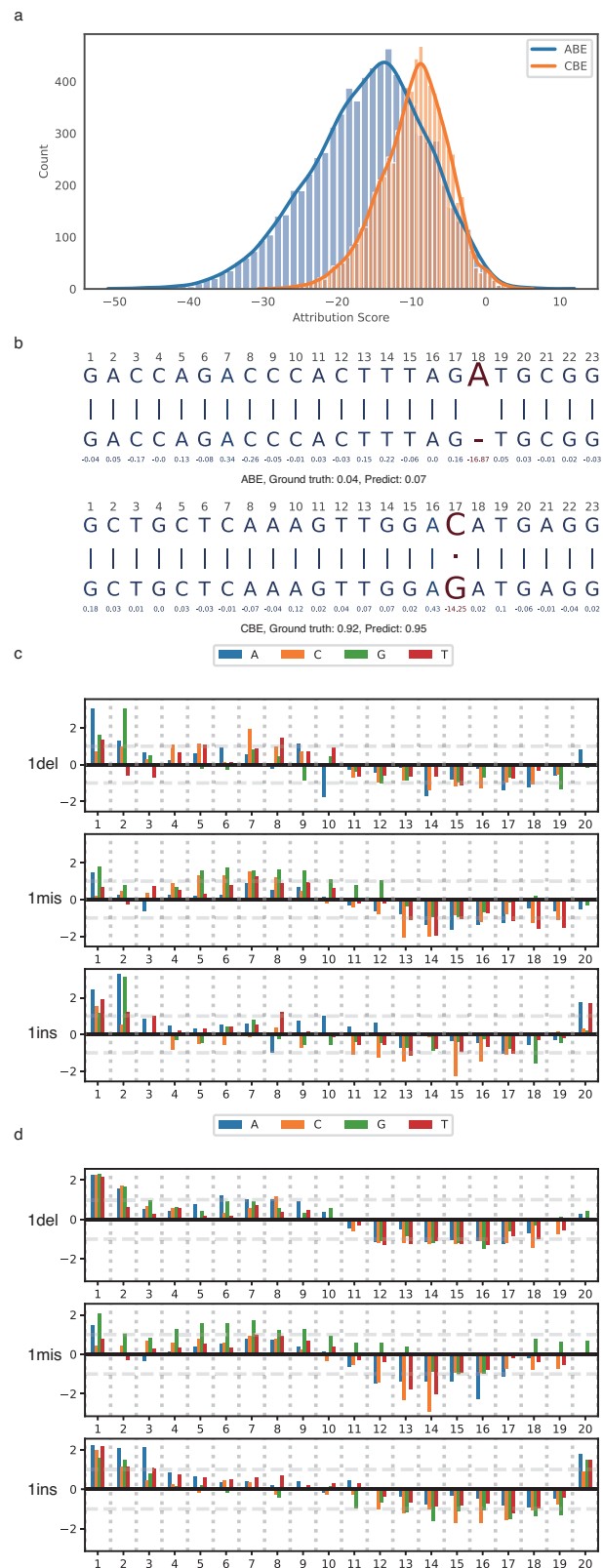

**Fig. 4 | Analysis of positional feature attributions associated with off:on-target ratio by LayerIntegratedGradient. a** Distribution of the attribution score for mutational nucleotides. **b** Two specific examples of feature attribution analysis. The mutational nucleotide alignment was marked in red. The numerical value below the nucleotide was the attribution score for that specific nucleotide position. The ground truth and predicted off:on-target ratio was also provided under the whole alignment sequence. **c**, **d** Positional feature attributions of off-target for 1 bp mismatch, 1 bp insertion, and 1 bp deletion on ABE and CBE, respectively. The attribution scores were standardized by the Z-score, where values higher than 1 or lower than −1 signified an important contribution. Source data are provided as a Source Data file.

hand-crafted feature algorithms, including Linear Regression, Ridge Regression, Multiple Perceptron, and XGBoost. The Spearman correlations of these models were lower than that of the CBEdeepoff model (Fig. 2c, Supplementary Data 9). These data demonstrated that the deep learning models outperformed the four conventional algorithms in the CBE dataset.

Next, we used CBEdeepoff to analyze different mutation types in the testing datasets. The model achieved very strong correlations for 1 bp deletion ($R = 0.887 \pm 0.015$), 2 bp mismatch ($R = 0.845 \pm 0.032$), and 1 bp insertion ($R = 0.811 \pm 0.022$); strong correlations for 1 bp mismatch ($R = 0.694 \pm 0.029$) and 2 bp insertion ($R = 0.689 \pm 0.077$); moderate correlations for 3 bp mismatches ($R = 0.575 \pm 0.116$), mixed mutations ($R = 0.549 \pm 0.188$), and for 2 bp deletions ($R = 0.478 \pm 0.075$); and very weak correlations for the remaining mutations (Fig. 3c). These results suggested that CBEdeepoff performed well for off-targets with 1–3 bp mismatches, 1–2 bp insertions, 1–2 bp deletions, and 2–3 bp mix mutations. Next, we evaluated the performance of the CBEdeepoff with six external groups of off-target datasets collected from the literature[19,20], where a series of 1–4 nt mismatched gRNAs were designed to target endogenous loci and achieved Spearman correlation scores varying from 0.710 to 0.859 (Fig. 3d, Supplementary Data 10).

Next, we evaluated our models with off-target datasets generated with a method called "Digenome-seq". This is a very sensitive in vitro method for genome-wide specificity detection of base editors[19,20]. We evaluated the performance of ABEdeepoff with 14 groups of off-target datasets generated by Kim et al.[20] and achieved a very strong correlation for gRNA 7 and a strong correlation for gRNA 8, mild correlations for four groups (gRNA 9-12), and weak or very weak correlations for the remaining eight groups (Supplementary Fig. 7, Supplementary Data 11). We evaluated the performance of CBEdeepoff with seven groups of off-target datasets generated by Kim et al.[20] and achieved strong correlation for gRNA 7, the mid correlation for gRNA 8, and weak or very weak correlations for the remaining groups (Supplementary Fig. 8, Supplementary Data 12). Our models performed poorly for these in external off-target datasets, probably due to the low editing efficiencies of the datasets. The majority of off-targets detected by Digenome-seq were 3–5 bp mismatches followed by mismatches plus a 1 bp deletion (Supplementary Fig. 9a, b). Other types of mutations were rarely observed. The editing efficiencies of these off-targets were similar to the background level except for 1-2 bp mismatched off-targets (Supplementary Figs. 9c, d).

## Model post-hoc explainability

To understand the feature contribution of the ABEdeepoff and CBEdeepoff models, we implement the "LayerIntegratedGradient" class from the Captum package[28] to the embedding layers to evaluate the attribution score for each nucleotide position in the input sequences. The attribution scores were calculated by taking the mean value of that position across the entire testing dataset. We found that attribution scores at mutation positions were all below zero, indicating that features contributed negatively to the off:on-target ratio (ABE = −15.96; CBE = −9.65, Fig. 4a). Attribution scores at matched positions were close to zero, indicating that features had minimal influence on the off:on-

target pairs in 1076 groups were randomly split into two parts by 10-fold "GroupKFold": one contained 51,123 pairs (968 groups), and the other contained 5,680 pairs (108 groups) for testing (Fig. 2b, Supplementary Fig. 6). The resulting model was named "CBEdeepoff", which can predict off:on-target ratios at potential off-targets. The model achieved a Spearman correlation of $0.863 \pm 0.012$ for the testing dataset. We also used the above training datasets to train conventional

target ratio (Fig. 4b). Furthermore, to estimate the contribution of the specific mutation types, we comprehensively analyzed the attribution score for 1mis, 1del, and 1ins for ABE and CBE. The attribution scores were standardized by the Z-score, where values higher than 1 or lower than −1 signified an important contribution. Overall, regardless of the mutation type, mutations at positions 1-10 had a relatively small impact on the off:on-target ratio (z-score above 0), while mutations at positions 10-20 usually had a large negative impact on the off:on-target ratio (Fig. 4c, d). However, for 1ins both in ABE and CBE, mutations at position 20 had a small impact on the off:on-target ratio.

We finally provided an online web server named BEdeepoff (http://www.deephf.com/#/bedeep/bedeepoff), which contains ABE-deepoff and CBEdeepoff for off-target prediction. Each model contains two modes. The text mode is for single off-target mutation prediction, and the file mode is for genome-wide off-target mutation prediction. For genome-wide off-target mutation prediction, users first use Cas-OFFinder (http://www.rgenome.net/cas-offinder)[29] or CRISPRitz[30] to identify DNA sequence similarity to the on-targets in the whole genome. We suggested setting up mismatches up to 3 nt, DNA bulge size 1nt, and RNA bulge size 1nt, which would cover all the high efficiency of off-targets. Next, the files generated from the suggested tools are uploaded to the ABEdeepoff/CBEdeepoff file mode, resulting in output files that contain the predicted off:on-target ratio for each sequence. One limitation of our study is that the gRNA-off-target pair library did not include sequences without editable nucleotides. In theory, the editing efficiency is 0 for such sequences. Therefore, the off:on-target ratio is set to 0 for sequences without editable nucleotides. In addition, the off:on-target ratio is set to 1 for sequences identical to the on-target sequence.

## Discussion

Off-target editing is always a major concern of base editing. As Cas9-deaminase fusion proteins, both ABE and CBE base editors can induce Cas9-dependent off-target mutations[18,19,31–33]. Eukaryote genomes contain hundreds of sequences similar to the target sequences and potentially be recognized by Cas9[34]. However, only a small portion of sequences can be edited by base editors. For a given target, it is crucial to know where and the efficiency of the off-target effect. In this study, we developed models to predict editing efficiencies on off-targets for both ABE and CBE, facilitating the evaluation of target specificity in silico. These models are particularly useful when a large number of potential gRNAs are available, for example, when generating stop codons to knock out a gene and designing gRNA libraries for knockout screens.

The fusion embedding-based deep learning models used here have a simple yet efficient structure that unifies on-target and off-target input. It can be seamlessly extended to other versions of base editors, such as SauriABEmax, SauriBE4max, SaKKH-BE3, BE4-CP, dCpf1-BE and eA3A-BE3[35–39], for off-target prediction.

## Methods

### Cell culture and transfection
HEK293T cells (ATCC) were maintained in Dulbecco's Modified Eagle Medium (DMEM) supplemented with 10% FBS (Gibco) at 37 °C and 5% CO2. All media contained 100 U/ml penicillin and 100 mg/ml streptomycin. For transfection, HEK293T cells were plated into 6-well plates, and DNA was mixed with Lipofectamine 2000 (Life Technologies) in Opti-MEM according to the manufacturer's instructions. Cells tested negative for mycoplasma.

### Plasmid construction
The sequences of the oligonucleotides are listed in Supplementary Data 13. The SB transposon (pT2-SV40-BSD-ABEmax and pT2-SV40-BSD-BE4max) was constructed as follows: first, we replaced the Neo^R gene (AvrII-KpnI site) on pT2-SV40-Neo^R with BSD, resulting in the pT2-

SV40-BSD vector; second, the backbone fragment of pT2-SV40-BSD was PCR-amplified with Gibson-SV40-F and Gibson-SV40-R, and the ABEmax fragment was PCR-amplified from pCMV_ABEmax_P2A_GFP (Addgene#112101) with Gibson-ABE/BE4-F and Gibson-ABE/BE4-R, and the BE4max fragment was PCR-amplified from pCMV_AncBE4max (Addgene#112094) with Gibson-ABE/BE4-F and Gibson-ABE/BE4-R; third, the backbone fragments were ligated with ABEmax and BE4max using Gibson Assembly (NEB), resulting in pT2-SV40-BSD-ABEmax and pT2-SV40-BSD-BE4max, respectively.

### Generation of cell lines expressing ABEmax or BE4max
HEK293T cells were seeded at ~40% confluency in a 6-well dish the day before transfection, and 2 μg of SB transposon (pT2-SV40-BSD-ABE-max or pT2-SV40-BSD- BE4max) and 0.5 μg of pCMV-SB100x were transfected using 5 μl of Lipofectamine 2000 (Life Technologies). After 24 h, cells were selected with 10 μg/ml blasticidin for ten days. Single cells were sorted into 96-well plates for colony formation. Conversion efficiency was performed to screen cell clones with high levels of ABEmax and BE4max expression.

### gRNA-target library construction
The off-target library was designed as follows: both ABEs and CBEs libraries contained 91,556 oligonucleotides, distributed among 1383 and 1378 gRNA groups for ABEs and CBEs, respectively; for ABEs, the off-target library included 51,170 1 bp-mismatch, 10,889 2 bp-mismatch, 10,310 1 bp-insertion, 4960 1 bp-deletion, 2261 3 bp-mismatch, 2264 4 bp-mismatch, 2255 5 bp-mismatch, 2260 6 bp-mismatch, 1086 2 bp-insertion, 1561 2 bp-deletion, 888 mix, 279 nontarget and 1383 on-target; for CBEs, the off-target library included 50,676 1 bp-mismatch, 11,054 2 bp-mismatch, 10,481 1 bp-insertion, 5128 1 bp-deletion, 2221 3 bp-mismatch, 2226 4 bp-mismatch, 2203 5 bp-mismatch, 2214 6 bp-mismatch, 1081 2 bp-insertion, 1631 2 bp-deletion, 881 mix, 392 non-target and 1378 on-target; each oligonucleotide chain contained left sequence (tgtggaaaggacgaaacacc), gRNA sequence (NNN NNNNNNNNNNNNNNNNNN), BsmBI site (gttttgagacg), Barcode 1 (NNNNNNNNNNNNNNNNNNNNN), BsmBI site (cgtctcgctcc), Barcode 2 (NNNNNNNNNNNNNNNN), target sequence (gtactNNNNNNNNNN NNNNNNNNNNNNNgg), and right sequence (cttggcgtaactagatct). The off-target library was constructed as follows: first, full-length oligonucleotides were PCR-amplified and cloned into the BsmBI site of the LentiGuide-U6-del-tracRNA vector by Gibson Assembly (NEB), named LentiGuide-U6-gRNA-target; second, the tracRNA was PCR-amplified and cloned into the BsmBI site of the LentiGuide-U6-del-tracrRNA vector by T4 DNA ligase (NEB). The Gibson Assembly products or T4 ligation products were electroporated into MegaX DH10B™ T1^R Electrocomp™ Cells (Invitrogen) using a GenePulser (BioRad) and grown at 32 °C and 225 rpm for 16 h. The plasmid DNA was extracted from bacterial cells using Endotoxin-Free Plasmid Maxiprep (Qiagen).

### Lentivirus production
Lentivirus production was described previously[21], briefly, for individual sgRNA packaging, 1.2 μg of gRNA-expressing plasmid, 0.9 μg of psPAX2, and 0.3 μg of pMD2. G (Addgene) was transfected into HEK293T cells by Lipofectamine 2000 (Life Technologies). The medium was changed 8 hours after transfection. After 48 h, virus supernatants were collected. For library packaging, 12 μg of plasmid library, 9 μg of psPAX2, and 3 μg of pMD2. G (Addgene) were transfected into 10-dish HEK293T cells with 60 μl of Lipofectamine 2000. Viruses were harvested twice at 48 h and 72 h posttransfection. The virus was concentrated using PEG8000 (no. LV810A-1, SBI, Palo Alto, CA), dissolved in PBS, and stored at −80 °C.

### Screening experiments in HEK293T cells
HEK293T cells expressing ABEmax or BE4max were plated into 15 cm dishes at ~30% confluence. After 24 h, cells were infected with

a library with at least 1000-fold coverage of each gRNA. After 24 h, the cells were cultured in media supplemented with 2 μg/ml puromycin for five days. Cells were harvested, and genomic DNA was isolated using Blood and cell Culture DNA Kits (Qiagen). The integrated region containing the gRNA coding sequences and target sequences was PCR-amplified using Q5 High-Fidelity 2X Master Mix (NEB). We performed 60-70 PCRs using 10 μg of genomic DNA as a template per reaction for deep sequencing analysis. The PCR conditions were 98 °C for 2 min, 25 cycles of 98 °C for 7 s, 67 °C for 15 s and 72 °C for 10 s, and a final extension at 72 °C for 2 min. The PCR products were mixed and purified using a Gel Extraction Kit (Qiagen). The purified products were sequenced on an Illumina HiSeq X by 150-bp paired-end sequencing.

## Data analysis
FASTQ raw sequencing reads were processed to identify gRNA-target editing activity. The nucleotides in a read with a quality score <10 were masked with the character "N". Due to the integrated design strategy, we first separated a read to the designed gRNA region, scaffold region, and target region to extract the corresponding sequence. The designed gRNA was then aligned to the reference gRNA library to mark the reads. A valid sequencing read must contain two designed barcode pairs. The target sequence was compared to the designed gRNA to mark if the target was edited. We screened out gRNAs with a total valid reading of less than 100. Then, the efficiency for a specific gRNA-target pair can be calculated by the following formula:

$$\text{editing efficiency} = \frac{\text{NO.of edited reads}}{\text{NO.of total valid reads}}$$

Further, in line with previous studies[26,27], to estimate the editing specificity (i.e., mutation tolerance) of a single-guide RNA, we also calculated the ratio of off-target efficiency to on-target efficiency (off:on-target ratio). The higher the off:on-target ratio, the lower the editing specificity. This metric made it possible to compare the specificity of all single-guide RNAs without considering the editing efficiency of their original matched target.

## Encoding
Drawing on concepts from the field of NLP, nucleotides A, C, G, and T can be regarded as words in a DNA sequence. Therefore, we can utilize the widely used algorithms in the NLP field to solve prediction tasks in the CRISPR field, especially embedding algorithms, to obtain the continuous representation of discrete nucleotide sequences[40]. Unlike the ordinary efficiency prediction that only needs to input one single sequence for regression models, in this research, the gRNA-off-target pair has two different sequences as inputs. For a gRNA, there are four words in the index vocabulary (i.e., A, C, G, and T); however, after alignment, there might exist a DNA bulge or RNA bulge and thus lead to a "-" word to represent the insertion or deletion in the gRNA or off-target sequence. Meanwhile, to be able to align sequences to the same length in every single batch, we add a <pad> token to the vocabulary:

$$D = \{0: <pad>, 1: A, 2: C, 3: G, 4: T, 5:-\}$$

The input sequence can be described as:

$$\boldsymbol{x_i} = \{x_{i0}, x_{i1} \cdots x_{it}, \cdots x_{i(T-1)}\},$$

where $i \in \{1,2\}$ denotes the $i$-th sequence in a gRNA-off-target pair, $x_{it}$ is the $t$-th element of the $i$-th sequence, and $T$ is the sequence length. For example, ACGCTTCATCA-ATGTTGGGATGG (seq1, gRNA + NGG)

and ACGC-TCATCAaAaGTT-GGATGG (seq2, off-target) can be encoded as:

$$\boldsymbol{x_1} = [1, 2, 3, 2, 4, 4, 2, 1, 4, 2, 1, \mathbf{5}, 1, 4, 3, 4, 4, 3, 3, 3, 1, 4, 3, 3] \text{ (i.e.,seq1)}$$

and

$$\boldsymbol{x_2} = [1, 2, 3, 2, \mathbf{5}, 4, 2, 1, 4, 2, 1, 1, 1, 1, 3, 4, 4, \mathbf{5}, 3, 3, 1, 4, 3, 3]$$
$$\text{(i.e.,seq2),respectively.}$$

## Embedding feature fusion
Inspired by the algorithms in the recommender system[41] and click-through rate (CTR)[42] prediction modeling, both the generalization capacity and training speed will benefit from the sharing of the same embedding matrix instead of training independent embedding matrices for each input. In this research, we use the same weight initialization settings instead of directly using the same embedding matrix and found it converged much faster. A discrete nucleotide encoding $x_{it}$ is projected to the dense real-valued space $\mathbf{E}_i \in \mathbb{R}^{T \times m}$ ($m$ is a hyperparameter corresponding to the embedding dimension) to obtain the embedding vector $\mathbf{e}(x_{it})$. Then, a final embedding matrix $\mathbf{E}$ is needed to obtain the combined information from those two embedding matrices by:

$$\mathbf{E} = g(\mathbf{E_1},\mathbf{E_2}) \tag{1}$$

where $g$ can be a sum, mean, or even a simple concatenate function. However, the sum or mean function is more suitable because it can reduce the redundant features in $\mathbf{E_1}$ and $\mathbf{E_2}$. We choose the sum function here for simplicity.

## Feature extraction and model prediction
The long short-term memory network (LSTM) and gated recurrent unit network (GRU) are recurrent neural network (RNN) algorithms used to address the vanishing gradient problem in modeling time-dependent and sequential data tasks[43]. Usually, a bidirectional manner is used to capture the information from the forward and backward directions of a sequence, which is biLSTM or biGRU. Our work and others' work have shown that, as an important component, biLSTM can be used alone or with a convolutional neural network (CNN) to achieve good performances in various regression and classification tasks involving biological sequences[21,44–46]. Here, we tested biLSTM, biGRU, and the newly proposed transformer structure[47] and found that biLSTM had the fastest convergence speed. The input and output of biLSTM can be described by the following equations:

$$\overrightarrow{\boldsymbol{h}_t} = \overrightarrow{LSTM}(\boldsymbol{e}(\overrightarrow{x}_t), \overrightarrow{\boldsymbol{h}}_{t-1}) \tag{2}$$

$$\overleftarrow{\boldsymbol{h}_t} = \overleftarrow{LSTM}(\boldsymbol{e}(\overleftarrow{x}_t), \overleftarrow{\boldsymbol{h}}_{t-1}) \tag{3}$$

Thus, the output context vectors of biLSTM are $\boldsymbol{h}_0 = [\overrightarrow{\boldsymbol{h}}_0; \overleftarrow{\boldsymbol{h}}_{L-1}]$, $\boldsymbol{h}_1 = [\overrightarrow{\boldsymbol{h}}_1; \overleftarrow{\boldsymbol{h}}_{L-2}]$, etc. Thus, we can concatenate the forward and backward hidden state as $\mathbf{H} = \{\boldsymbol{h}_0, \boldsymbol{h}_1, , \boldsymbol{h}_{L-1}\}$, which contains the bidirectional information in the embedding feature matrix. Before the fully connected layers, we tried different input features based on the trade-off of the convergence speed and the performance of the model. The aforementioned features are the last hidden unit, max pooling operation on $\mathbf{H}$ and attention pooling on $\mathbf{H}$. The equations are as

follows:

$$\boldsymbol{h}_{Last} = [\vec{\boldsymbol{h}}_{L-1}; \overleftarrow{\boldsymbol{h}}_{L-1}] \tag{4}$$

$$\mathbf{F}_{MaxPool} = \max_{t=1}^{L-1} \boldsymbol{h}_t \tag{5}$$

$$\mathbf{F}_{AttentionPool} = \text{attention}_{t=1}^{L-1} \boldsymbol{h}_t \tag{6}$$

A max pooling and attention pooling function were applied separately to **H** to obtain more useful features and combine them with the final hidden state to produce the concatenated feature $\boldsymbol{c} = [\boldsymbol{h}_{Last}; \mathbf{F}_{MaxPool}; \mathbf{F}_{AttentionPool}]$. The predicted off:on-target ratio $o$ for a specific gRNA-off-target pair can be obtained by:

$$o = \sigma(f(\boldsymbol{c})) \tag{7}$$

where $f$ are fully connected layers and $\sigma$ is the *sigmoid* activation function.

Considering that the sample sizes of off-target types vary greatly, to be able to dynamically adjust the loss weight of off type in the training data, we use the inverse ratio of prevalence of each category in each batch to weight the Mean Squared Error of corresponding mutation type.

### Training setting

To avoid overfitting, off-target datasets were grouped by on-target sequence to get different gRNA groups, and these groups were randomly divided into training and testing datasets, with 90% of the gRNA groups assigned to the training set and 10% to the testing set. The stability of model performance was estimated by a 10-fold "GroupK-Fold" validation together with the external off-target datasets. The ABEdeepoff and CBEdeepoff models share the following hyperparameters: embedding dimension, 256; LSTM hidden units, 512; LSTM hidden layers, 2; dropout rate, 0.5; fully connected layers, 2 (6*512->3*512->1).

For all the models, the Adam optimizer was used with a customized learning rate decay strategy that gradually reduced the learning rate from 0.001, 0.0001, and 0.00001 to 0.000005.

### Comparison with baseline models

We compared four conventional hand-crafted feature algorithms: Linear Regression, Ridge Regression, Multiple Perceptron, and XGBoost with the proposed deep learning methods ABEdeepoff and CBEdeepoff, which are primarily constructed using LSTM. The features used in the conventional algorithms were 1-mer position-dependent nucleotide composition, 1-mer position-independent nucleotide composition, GC content, and RNA/DNA binding free-energy[48,49]. The nearest neighbor method was used to calculate the binding free energy: a 2 bp sliding window traverses the matching part of the off-target sequence when aligned with the gRNA, every two bases in the sliding window correspond to a unique energy, and the sum of all window energy is the final energy used in the model. We use a Bayesian optimization method named TPE (Tree-structured Parzen Estimator) provided by Optuna[50] package to obtain the optimal hyperparameters (Supplementary Data 14).

### Tools used in the study

BWA-0.7.17 was used to identify the designed gRNA[51]. The global algorithm of pairwise2 in BioPython 1.78[52] was used to obtain the alignment result of the off-target sequence. Scikit-learn 1.1.1[53] and PyTorch 1.13.1[54] were used for conventional machine-learning algorithms and deep-learning models, respectively.

### Statistics and reproducibility

All the data are shown as the mean ± S.D. The sample size were determined by the limited library size. These gRNAs were randomly selected from our previously designed gRNA library. The gRNA with total reads less than 100 were removed. The testing datasets were randomly selected.

### Reporting summary

Further information on research design is available in the Nature Portfolio Reporting Summary linked to this article.

## Data availability

The editing efficiency and off:on-target ratio of the entire dataset can be found in Supplementary Data 1-2. The raw sequencing data has been submitted to the NCBI Sequence Read Archive SRA PRJNA587328. Source Data is available as a Source Data file. Source data are provided with this paper.

## Code availability

All software codes of the study are available on GitHub (https://github.com/izhangcd/BEdeep)[55]. We have integrated the model (trained with the full dataset instead of leaving a holdout testing) into an online web server (http://www.deephf.com/#/bedeep/bedeepoff). One can input gRNA and off-target sequence pairs to predict the off:on-target ratio.

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

## Acknowledgements

This work was supported by grants from the National Key Research and Development Program of China (2021YFC2701103, 2021YFA0910602, 2022YFD2101500); the National Natural Science Foundation of China (82070258, 82030099); the Open Research Fund of State Key Laboratory of Genetic Engineering, Fudan University (No. SKLGE-2104); and the Science and Technology Research Program of Shanghai (19DZ2282100, 22DZ2303000); Central Guidance on Local Science and Technology Development Fund of Hubei Province (2022BGE270).

## Author contributions

C.-D.Z. and Y.Y. analyzed the data, built the prediction model, and created the website. Q.T., L.-H.H., and J.-J.W. performed experiments. Y.-N.Z. and J.-C.Y. analyzed the data. L.-M.S. supervised the data analysis and revised the paper. S.-G.O., H.-Y.W., and H.W. supervised the project. B.Y. and Y.-M.W. designed the project and wrote the paper. All authors read and approved the final paper.

## Competing interests

The authors declare no competing interests.
