## [Peer Review File · Nature Communications]

REVIEWER COMMENTS

Reviewer #1 (Remarks to the Author):

In this study, Zhang and colleagues proposed two shared-embedding and biLSTM-based deep learning models to predict off-target editing efficiency for CBE and ABE respectively. Firstly, the authors performed ABE and CBE large-scale screens with more than 90,000 gRNA-on/off-target pairs containing up to six mismatches in ABE and CBE stably expressed HEK293T cells. The gRNA-target sequence was amplified from genomic DNA five days after transduction and subsequently sequenced. The conversion/editing efficiency, defined as the ratio of the number of edited reads and total reads, of the resulting less than 60,000 gRNA-target pairs was used to investigate the effect of the number and the position of mismatches and to train two deep learning models. The model performance was evaluated with 10-fold cross-validation and independent off-target datasets from previous studies and generated with Digenome-Seq, resulting in a wide range of case-dependent accuracy.

I think it is of significance to evaluate the off-target effect with a specified and comprehensive dataset, and the predictive model will ease the application of the base editor. That said, I have some comments on the manuscript:

Major comments:

1. Line 71 & 237 in the main text and method section, the authors only mentioned the number of the groups of gRNAs but didn't describe how to select or design these groups. It would be worth knowing the rationale for gRNA design, the selection criteria, and how well these groups represent the potential gRNAs in the human genome, such as GC content, positional base preference, nucleotide frequency, etc.
2. Figure 1c-f, the authors showed that the efficiency of both on-targets and off-targets varied dramatically and there were positional effects of mismatches. It would be nice to combine the effect of the on-targets to evaluate the interaction effects between gRNA: DNA pairing and base editing considering the editing efficiency of on-targets could drop to 13.7%. I am concerned that the lower on-target editing efficiency might confound the efficiency shown in e and f. Moreover, in e and f, it would be nice to know if there are significant differences across positions.
3. In Figures 3a&c, the authors showed the model performance of each mismatch group. Although the authors applied weighted MSE for the unbalanced sample size, the performance in groups with more than 4 mismatches was still much worse. Did the authors try to improve the generality of the model? Could the model perform better if the convergence speed was not prioritized as mentioned in lines 357 and 365?
4. Figure 3b&d, the models are validated with independent data from previous studies. There could potentially be examples of Simpson's paradox here which can cause misleading correlations in some cases, such as gRNA2-5 in b and gRNA3-6 in d, due to an excessive number of values around zero in the

measured efficiencies. I think it is worth calculating the correlation after subsetting the data points and interpreting how the model outputs a much higher predicted efficiency compared to the measured efficiency.

Minor comments:

1. Line 40, “~100 nt guide RNA”. Do the authors mean sgRNA instead? Because the guide RNA usually means the sequence that matches the target DNA, it might be confusing to call the 100 nt gRNA-repeat sequences gRNA.
2. In line 48, the authors mentioned prokaryotes and plants as examples but the references do not.
3. In Line 96, when the author said “Data from the two replicates were combined for subsequent analysis”, what does it mean exactly? It is also not mentioned in the methods.
4. In lines 98 and 101, only ~55000 gRNA-target pairs remained for the following analysis. What caused the loss of over one-third of the library?
5. In Line 121, the seed region was never specified in the text, so it might be more informative to include the definition of the seed regions.
6. Line 276, could the author specify the “suitable time points”?
7. The resolution of the figures is a bit low, especially in figure 2.

Reviewer #2 (Remarks to the Author):

Zhang et al developed prediction models for off-target activity of base editors. For each type of base editor, converting C->T or A->G respectively, activity of a large number of guide RNAs mismatched to off-targets (ca. 55 000) was obtained by NGS and the resulting datasets were used to train a deep learning model. Prediction of off-target activity of base editors is an important topic in the field of gene editing. In my opinion, however, several aspects of the work need to be strengthened or clarified.

Major comments

- 1) The approach followed by the authors to devise the prediction tools is now well established and is based on statistical analysis by AI methods of a large dataset generated with a library of lentiviral vectors. Validation of prediction models was done on previously published datasets of base editor off-targets and good correlations to experimental results were achieved, indicating good performance of the prediction models. I don't have the skills to evaluate if the statistical analysis was optimal and whether the predictions achieved could be further improved. Nevertheless, other methods of statistical might be possible and could prove beneficial.

2) The applications of base editing for which prediction models are useful should be discussed. Base editing has been developed foremost for gene therapy. For correction of specific mutations by base editing, only very few guides can be used, if any (because base editors are only efficient at converting nucleobases in a small deamination window of the guide RNA target sequence). In this case, it is generally straightforward to experimentally check for levels of unwanted editing at the limited number of potential off-target sites identified *in silico*. In contrast, the prediction models developed by Zhang et al will be useful for applications of base editing where a large number of potential guides can be used, for example when generating stop codons and designing guide libraries for knock-out screens.

3) One well-known determinant of editing efficiency is the position of target nucleobases in the deamination window. This parameter should be taken into account when predicting editing efficiency at off-targets (for example using models of editing efficiency at the different positions in the base editing window) (or better highlighted in the manuscript if this was already done!).

Minor comments

1) The number of mismatched gRNA-targets for which base editing activity was obtained was significantly lower than the number of gRNA-targets tested (e.g. 54,663 out of 91,287 for ABE). Could the authors comment on that difference and whether it may have affected their dataset and modelling?

2) As indicated above, good correlations to experimental results were achieved, indicating good performance of the prediction models. For potential off-targets with higher number of mismatches (≥ 3), however, correlations were lower. Is it correct that predictions generally overestimated editing at such potential off-targets? The experimental system used by the authors, i.e. a cell line stably expressing base editors and giving remarkably high on-target base editing, may have favored unusually high activity at off-targets with multiple mismatches and relatively poor alignment with the guide sequence and thus biased their predictions.

3) The guides shown in Figure 1e both have two target nucleobases in the “base editing window”. This seems to have been designed on purpose and should be explained.

4) The online tool for prediction of off-target editing is very simple and nice to use. It would perhaps benefit from including off-target search without having to visit the Cas-OFF or Caspritz websites.

Figure 1b The reason for selecting the edits represented in Fig. 1b is not clear and editing efficiency is not given. Please clarify and indicate the corresponding editing efficiencies.

Figure 1e and 1f The finding that a 1bp mismatch or a 1 bp insertion at position 20 has almost no impact on base editing activity is totally unexpected. The authors should check the impact of these specific mutations on Cas9 cleavage activity for some of the sequences tested.

Page 14 Line 280 The method for quantifying conversion efficiencies from Sanger sequencing files should be detailed.

Page 6 Line 121 Give the exact definition of the seed region used for the analysis.

Page 2 Line 50 Typo correction needed

Reviewer #3 (Remarks to the Author):

In this work, the authors report on their study of designing gRNA mismatched target libraries and consequently training machine learning models (neural network based model) to predict editing efficiency at off-targets for both (1) Adenine base editor (ABEs) and (2) Cytosine base editors (CBE). The authors reported on the performance of their trained models on held-out test sets (test sets from the 10 times shuffle split of the data) and on assembled data from the literature (i.e. external data). Overall the reported models achieved good performance on the generated experimental data (i.e. in-house data) and modest to low performance on external data. The authors provide access to the models through public code repository and a web app that accepts a gRNA-off-target pair and outputs the predicted editing efficiency.

While reading the manuscript and inspecting the code, I had couple of comments and suggestions:

1- one main issue in the setup is the data splitting. The authors considers the whole set of sequence pairs (on- and off- target) as the unique “unit of analysis” and then proceed to do splitting of data into training and testing. The problem is that the data is based on pairs from the gRNA-on-target pair (i.e. initial sequences) and then multiple gRNA-off-target pairs associated with them. For example, in case of ABE data, you have 1110 unique sequences (on-target sequences) and 54663 off-target sequences associated with these initial sequences. In a trivial way, when splitting the data we might end up having pairs of same initial sequence ending up in both training and testing. This design would lead to a “leak” as both sequence pairs are correlated — this is amplified given that major off-targets are based on 1bp mismatch, insertion or deletion. To test this hypothesis, if we group the data by these 1110 sequences (i.e. the “source” column in the ABE data frame), and then observe the average editing efficiency within each group, the multiple off-targets are generally close to that mean.

Hence, a more unbiased option is to (1) first group the sequences by the “on-target” sequences (in case of ABE, it would be the 1110 unique sequences in the “source” column) and then perform the data splitting based on these sequences (i.e. the group number associated to these sequences). In other words, we would have 1110 ids to distribute between training or testing and then all off-target pairs associated with these ids could be used safely as there is no chance of having same initial (on-target) associated to train and test sets.

1.1 A related comment to the splitting setup is to use 5-fold or 10-fold cross-validation rather than “shuffle split”. As mentioned in point 1, after the grouping and doing the splitting based on the unique “source” (on-target) sequences, I suggest to do 5-fold or 10-fold splitting. The reason is that with this setup, all sequence pairs will be in the test set at one point in the evaluation. Whereas the shuffle split does not guarantee that and also does not guarantee that all folds will be different (please see here https://scikit-learn.org/stable/modules/generated/sklearn.model_selection.ShuffleSplit.html). To conclude this point, my suggestion is to use some variation of GroupKFold (https://scikit-learn.org/stable/modules/generated/sklearn.model_selection.GroupKFold.html) where the grouping is based on the unique on-target sequences (i.e. 1110 in the case of ABE).

1.2 Another comment related to the reporting in the writeup about the splitting procedure. The authors refer to the shuffle split as “cross-validation” see page 6 line 131 “.. for tenfold cross-validation” and similarly page 8 line 147. If the authors would continue with the shuffle split approach, I think rewording these sentences would be more accurate.

2. The authors reported on the design of neural network based model that uses shared embedding matrix and bidirectional RNN layer followed by output regression layer. The design space for the neural network models (i.e. choice of architecture) is huge and there are many variations of computational layers that could be used for modelling the current problem. I think it is unfair to ask for different models (i.e. designs) based on other architectures etc. But I think it is helpful to assess to what degree the neural model (current architecture) is successful in solving the prediction problem relative to a baseline approach. And this could be done comparing it to a baseline to understand the difficulty of the prediction task. For example fitting a gradient boosting based model (such as Xgboost) would be a great baseline to compare the added value of the current neural network architecture.

3. Looking at Figure 1 C and D, it can be concluded that On, 1mis, 1ins, 1del cases have wider variation of editing efficiency compared to other off-target mutations that cluster at the low editing rate. It would be interesting to inspect the models predictions and highlight what learned features (i.e. position and nucleotides) for a given pair that contribute to the model’s decision of increased or decreased editing efficiency prediction. Currently we have descriptive statistics (i.e. distribution) on the different editing rates but not much on the models’ prediction process.

4. When reporting the models performance on experimental test sets, we see average performance of the 10 folds trained models with the standard deviation (Figure 3 A and C). But when it comes to external data, it is not clear if the performance is based on (a) one specific chosen model, or (b) it is the average of predictions from the 10 models (such as mean prediction) correlated with the ground-truth, or (c) average performance across the 10 fold models. Maybe captioning the figure with this info (or mentioning it in the main manuscript) is helpful for reproducibility.

5. Some inconsistencies noted between the manuscript and the code repo shared.

5.1 I noted that during training the batches are not shuffled with every iteration see <https://github.com/izhangcd/BEdeep/blob/767082f0e05488b228783581eb900b3c7a16e5fb/training.py#L350> I think this might be a bug as you typically would like to shuffle the mini batches during training

5.2 For the embedding of sequence pair (i.e. gRNA and off-target), equation 1 (line 341), the authors refer to using the “sum” of both embedding at each position. It might be worth mentioning that `nn.EmbeddingBag` (<https://pytorch.org/docs/stable/generated/torch.nn.EmbeddingBag.html>) used in the code <https://github.com/izhangcd/BEdeep/blob/767082f0e05488b228783581eb900b3c7a16e5fb/training.py#L114> by default uses “mode=mean” (that is it averages rather than summing unless explicitly specified).

5.3 I was wondering about the weights used while computing the loss function. They are specific to each “off type” in the data <https://github.com/izhangcd/BEdeep/blob/767082f0e05488b228783581eb900b3c7a16e5fb/training.py#L24> Generally, you would estimate these from the training data (i.e. training partition) which I assume using inverse ratio of prevalence of each category. But it seems they are given and most probably estimated using the whole data. I do not think this will affect the results or the dynamics of the training but I bring the authors’ attention to this point.

6. General comments on the figures:

6.1 For figure 1 C and D, it might be helpful to add number of samples on the margin/side (i.e. $n=xx$) to basically complete the picture of how many samples for each distribution - given that there is a huge difference between the categories (i.e. >33000 1mis and the second next category is >6000 for 2mis, etc..)

6.2 The model architecture schematic (Figure 2 A) could become clearer if either the input is (1) one sequence pair or (2) batch sequence pairs (as it is now) but the shared embedding would have a stacked representation (i.e. overlaid) of the different sequence pairs. I say this because it was confusing to follow as it seems the shared embedding part refers to the embedding of each position in one sequence pair rather a batch of sequence pairs — I might be wrong interpreting it so I apologise in advance.

6.3 I was wondering in Figure 1 e and f, why does the editing efficiency bounces back (it increases) when it is at position 20? I know it is near the PAM, but looking at the previously cited paper (ref [19]), it shows that nucleotide at position 20 (typically G) leads to higher editing for a sequence. Does that mean, the 1mis would be having a letter switching to G in these cases, or when it is 1ins it is a G inserted at that position?

NCOMMS-22-37542-T (Prediction of base editor off-target activity by deep learning)

We sincerely thank the reviewers for their positive comments and constructive suggestions to improve our manuscript. A copy of the comments (highlighted in grey), followed by a detailed point-by-point response, can be found below. All changes and additions to the manuscript are marked for ease of review.

REPLY TO REFEREE #1:

In this study, Zhang and colleagues proposed two shared-embedding and biLSTM-based deep learning models to predict off-target editing efficiency for CBE and ABE respectively. Firstly, the authors performed ABE and CBE large-scale screens with more than 90,000 gRNA-on/off-target pairs containing up to six mismatches in ABE and CBE stably expressed HEK293T cells. The gRNA-target sequence was amplified from genomic DNA five days after transduction and subsequently sequenced. The conversion/editing efficiency, defined as the ratio of the number of edited reads and total reads, of the resulting less than 60,000 gRNA-target pairs was used to investigate the effect of the number and the position of mismatches and to train two deep learning models. The model performance was evaluated with 10-fold cross-validation and independent off-target datasets from previous studies and generated with Digenome-Seq, resulting in a wide range of case-dependent accuracy.

I think it is of significance to evaluate the off-target effect with a specified and comprehensive dataset, and the predictive model will ease the application of the base editor. That said, I have some comments on the manuscript:

Major comments:

1. Line 71 & 237 in the main text and method section, the authors only mentioned the number of the groups of gRNAs but didn't describe how to select or design these groups. It would be worth knowing the rationale for gRNA design, the selection criteria, and how well these groups represent the potential gRNAs in the human genome, such as GC content, positional base preference, nucleotide frequency, etc.

Respond:

Thank you for your valuable comments to improve our manuscript. These gRNAs were randomly selected from our previously designed gRNA library targeting human genome

1. The GC content of the ABE library accounted for 52.99%; the GC content of the CBE library accounted for 55.15%. The GC content of the ABE library positionally varied from 38.83 at position 7 to 65.80% at position 20; the GC content of the CBE library positionally varied from 43.40% at position 14 to 66.04% at position 20 (Supplementary Tables 3 and 4). We have added this statement in the first paragraph of the Results.

2. Figure 1c-f, the authors showed that the efficiency of both on-targets and off-targets varied dramatically and there were positional effects of mismatches. It would be nice to combine the effect of the on-targets to evaluate the interaction effects between gRNA: DNA pairing and base editing considering the editing efficiency of on-targets could drop to 13.7%. I am concerned that the lower on-target editing efficiency might confound the efficiency shown in e and f. Moreover, in e and f, it would be nice to know if there are significant differences across positions.

Respond:

Thank you very much for your advice. In the revised manuscript, the off-target editing efficiencies were normalized by their corresponding on-target efficiencies (off:on-target ratio) for the research throughout the manuscript. All graphs and tables have been recreated, and the models are also re-trained based on the off:on ratios.

In Supplementary Tables 12 and 13, we used an independent T-test to compare the off:on-target ratio across different positions, and a Bonferroni correction for multiple tests was applied. It's obvious that the statistical results (p-value) indicate significant differences in most positions within the 1mis, 1ins, and 1del subgroups.

3. In Figures 3a&c, the authors showed the model performance of each mismatch group. Although the authors applied weighted MSE for the unbalanced sample size, the performance in groups with more than 4 mismatches was still much worse. Did the authors try to improve the generality of the model? Could the model perform better if the convergence speed was not prioritized as mentioned in lines 357 and 365?

Respond:

Thank you for your comments. We think two main factors cause the lack of prediction capability for more than 4 mismatches. Firstly, there is the issue of data size, as the combination situations of more than 4 mismatches is very large, and we can only randomly cover a small portion of these combinations. Secondly, the editing efficiency of off-targets with more than 4 mismatches is very low and centralized (Revision Figure 1), which makes it difficult for the model to distinguish them.

Revision Figure 1. The editing efficiency and off:on-target ratio distribution of 4-6 mismatches.

To improve the model's generalization, we employed a stepped learning rate decay strategy during the training process (from 0.001, 0.0001, 0.00001 to 0.000005). This approach automatically adjusts the learning rate to a lower value when the current learning rate fails to converge further in more than 4 epochs. We found this strategy to be effective during the training process. Besides, in the revised manuscript, we changed the optimizer from Adam to Adamw². The latter is an improvement over Adam that adds L2 regularization to the cost function and adjusts the learning rate in a slightly different way. L2 regularization can effectively mitigate overfitting while improving the performance of Adam when dealing with sparse gradients (frequently appearing in sequence-like problems).

4. Figure 3b&d, the models are validated with independent data from previous studies. There could potentially be examples of Simpson's paradox here which can cause misleading correlations in some cases, such as gRNA2-5 in b and gRNA3-6 in d, due to an excessive number of values around zero in the measured efficiencies. I think it is worth calculating the correlation after subsetting the data points and interpreting how the model outputs a much higher predicted efficiency compared to the measured efficiency.

Respond:

Following the reviewer's suggestion, we applied a filter requiring efficiencies greater than 1%, 10%, and 20%, respectively, while also ensuring that the number of remaining gRNAs was greater than 15. We recalculated the Spearman correlation scores after applying these filters and found that there was still a middle to high correlation between the actual and predicted off:on-target ratio in different efficiency range subsets (Supplementary Tables 14 and 15). Our goal was to be able to assess the effect of mutations in off-target sequences, so as mentioned in the paper, we chose efficient clones for high-throughput experiments. The overall high editing efficiency in our library is the reason why the predicted editing efficiency is also high. However, when we utilize the off:on-target ratio, the editing efficiency was normalized by its corresponding on-target efficiency, which will make it comparable across the different original on-target efficiencies.

Supplementary Table 14. The third-party ABE off-target datasets prediction results after selection.

Efficiency threshold	Group	Number	Spearman correlation
--------------	---------------	-----------------------------

1%	gRNA_3	29	0.8203
1%	gRNA_2	22	0.7644
1%	gRNA_6	35	0.7462
1%	gRNA_1	22	0.6838
1%	gRNA_5	24	0.4304
10%	gRNA_3	22	0.6104
10%	gRNA_1	16	0.3794
10%	gRNA_6	21	0.2909
20%	gRNA_3	21	0.5519

Supplementary Table 15. The third-party CBE off-target datasets prediction results after selection.

Efficiency threshold	Group	Number	Spearman correlation
1%	gRNA_5	26	0.7149
1%	gRNA_2	18	0.6945
1%	gRNA_1	19	0.6439
1%	gRNA_3	17	0.4730

1%	gRNA_4	15	0.3357
10%	gRNA_1	15	0.7214
10%	gRNA_5	21	0.6935
10%	gRNA_3	15	0.4821
20%	gRNA_5	18	0.5769

Minor comments:

1. Line 40, "~100 nt guide RNA". Do the authors mean sgRNA instead? Because the guide RNA usually means the sequence that matches the target DNA, it might be confusing to call the 100 nt gRNA-repeat sequences gRNA.

Respond:

Thank you for your suggestions. In the revised manuscript, we stated, "Cas9^{D10A} nuclease and a ~100 nt single-guide RNA (sgRNA) form a Cas9^{D10A}-sgRNA complex".

2. In line 48, the authors mentioned prokaryotes and plants as examples but the references do not.

Respond:

Thank you for your comments. References have been added to the revised manuscript.

3. In Line 96, when the author said "Data from the two replicates were combined for subsequent analysis", what does it mean exactly? It is also not mentioned in the methods.

Respond:

Supplementary Figure 2 illustrates the flowchart of the data preparation strategy. As described in the manuscript, we first calculated the editing efficiency between the two technical replicates. The high Pearson correlation score between them (0.970 for ABE and 0.994 for CBE) was the basis for the merging of the two replicates. Taking ABE as an example, we first obtained the intersection set of two replicates and then computed the average editing efficiency for the same gRNA in the intersection set. For example, if an off-target efficiency was 0.5 in replicate 1 and 0.7 in replicate 2, the final off-target efficiency was 0.6. For unique gRNAs, the editing efficiency of the different sets of Replicate A and the different sets of Replicate B was directly used as the final editing efficiency. Reads for gRNAs less than 100 were also removed. This description was added to the Methods and Results parts of the manuscript.

4. In lines 98 and 101, only ~55000 gRNA-target pairs remained for the following analysis. What caused the loss of over one-third of the library?

Respond:

As illustrated in the above respond, the high-throughput sequencing data from replicate 1 and 2 identified 79,997 (87.6%) gRNA-target pairs for ABE and 76,106 (83.5%) gRNA-target pairs for CBE. In order to avoid unprecise efficiency, we removed about 24,224 (26.5%) and 19,303 (21.2%) pairs with read counts less than 100, leaving approximately 55,773 (61.1%) pairs for ABE and 56,803 (62.3%) pairs for CBE.

5. In Line 121, the seed region was never specified in the text, so it might be more informative to include the definition of the seed regions.

Respond:

Thank you for pointing this out. The seed region is the region where 1~9 nucleotides proximal to the PAM³⁻⁶. We added this explanation in the last paragraph of the first section of the results.

6. Line 276, could the author specify the "suitable time points"?

Respond:

"suitable time points" was replaced by "day five" in the revised manuscript.

7. The resolution of the figures is a bit low, especially in figure 2.

Respond:

We have enhanced the resolution of all figures.

REPLY TO REFEREE #2:

Zhang et al developed prediction models for off-target activity of base editors. For each type of base editor, converting C->T or A->G respectively, activity of a large number of guide RNAs mismatched to off-targets (ca. 55 000) was obtained by NGS and the resulting datasets were used to train a deep learning model. Prediction of off-target activity of base editors is an important topic in the field of gene editing. In my opinion, however, several aspects of the work need to be strengthened or clarified.

Major comments

1) The approach followed by the authors to devise the prediction tools is now well established and is based on statistical analysis by AI methods of a large dataset generated with a library of lentiviral vectors. Validation of prediction models was done on previously published datasets of base editor off-targets and good correlations to experimental results were achieved, indicating good performance of the prediction models. I don't have the skills to evaluate if the statistical analysis was optimal and whether the predictions achieved could be further improved. Nevertheless, other methods of statistical might be possible and could prove beneficial.

Respond:

Thank you for your comments. Figure 2b displays the schematic of the dataset and algorithms used for benchmarking the off-target predicting task. We compared four conventional hand-crafted feature algorithms: Linear Regression, Ridge Regression, Multiple Perceptron, and XGBoost with the proposed deep learning methods ABEdeepoff and CBEdeepoff, which are primarily constructed using LSTM. Due to the superior feature extraction and representation capabilities as well as the model complexity, the LSTM backbone used in our deep learning models for deep learning significantly outperforms the other four traditional algorithms. Besides, we should mention that based on the Reviewer 1's comments, it's more appropriate to use the ratio of off-target efficiency to on-target efficiency (off:on-target ratio) in our study than to use editing efficiency directly, which would have reduced the impact of a wide distribution of on-target edit efficiency on

off-target assessment. Supplementary Table 5. The benchmark results of various models for ABE.

Algorithm	Spearman correlation Mean	Spearman correlation STD
Linear	0.350	0.099
Ridge	0.352	0.095
XGBoost	0.481	0.067
MLP	0.524	0.057
LSTM	0.787	0.004

Supplementary Table 7. The benchmark results of various models for CBE

Algorithm	Spearman correlation Mean	Spearman correlation STD
Linear	0.389	0.045
Ridge	0.395	0.044
XGBoost	0.592	0.053
MLP	0.571	0.044
LSTM	0.855	0.003

2) The applications of base editing for which prediction models are useful should be discussed. Base editing has been developed foremost for gene therapy. For correction of specific mutations by base editing, only very few guide can be used, if any (because base editors are only efficient at converting nucleobases in a small deamination window of the guide RNA target sequence). In this case, it is generally straightforward to experimentally check for levels of unwanted editing at the limited number of potential off-target sites identified in silico. In contrast, the prediction models developed by Zhang et al will be useful for applications of base editing where a large number of potential guides can be used, for example when generating stop codons and designing guide libraries for knock-out screens.

Respond:

Thank you for your constructive comments. We discussed these applications in the revised manuscript.

3) One well-known determinant of editing efficiency is the position of target nucleobases in the deamination window. This parameter should be taken into account when predicting editing efficiency at off-targets (for example using models of editing efficiency at the different positions in the base editing window) (or better highlighted in the manuscript if this was already done!).

Respond:

Thank you for your comment. Since the input of the deep learning model in our study are sequences of off-target, and it's matched on-target, it preserves the original order features

of the nucleotides, including the base editing window. The deep learning model can automatically learn the important position-dependent nucleotide features in a way humans can or cannot understand. However, we used a method called "LayerIntegratedGradients" to understand the importance of the input nucleotides. We found that, in general, mutation at any position negatively affects editing efficiency, but mutations within the deamidation window have a lesser impact on the final editing efficiency compared to mutations at the nucleotides proximal to the PAM, which is typically considered an important region for RNA:DNA binding. We have added a section named "The interpretability of the models" in the Results part to discuss this.

Minor comments

1) The number of mismatched gRNA-targets for which base editing activity was obtained was significantly lower than the number of gRNA-targets tested (e.g. 54,663 out of 91,287 for ABE). Could the authors comment on that difference and whether it may have affected their dataset and modelling?

Respond:

As illustrated in Supplementary Figure 2, the high-throughput sequencing data from replicate 1 and 2 identified 79,997 (87.6%) gRNA-target pairs for ABE and 76,106 (83.5%) gRNA-target pairs for CBE. In order to avoid unprecise efficiency, we removed about 24,224 (26.5%) and 19,303 (21.2%) pairs with read counts less than 100, leaving approximately 55,773 (61.1%) pairs for ABE and 56,803 (62.3%) pairs for CBE. Based on results from both internal and external testing datasets, it was found that the model built using 55,000-57,000 gRNA-target pairs exhibited robust performance. Therefore, the

remaining filtered data did not significantly impact the model's performance. However, it should be noted that filtering too many invalid sequences may lead to a reduction in off-target designs such as 4, 5, 6mis, 2del, 2ins, etc. This reduction may result in the model's prediction ability on such off-targets being inferior to that on 1ins, 1del, and 1mis.

2) As indicated above, good correlations to experimental results were achieved, indicating good performance of the prediction models. For potential off-targets with higher number of mismatches (≥ 3), however, correlations were lower. Is it correct that predictions generally overestimated editing at such potential off-targets? The experimental system used by the authors, i.e. a cell line stably expressing base editors and giving remarkably high on-target base editing, may have favored unusually high activity at off-targets with multiple mismatches and relatively poor alignment with the guide sequence and thus biased their predictions.

Respond:

As respond to the first major comment, Reviewer 1 suggested us to normalize the off-target editing efficiency with the on-target efficiency to get a new metric named "off:on-target ratio", which will make it comparable across the different original on-target efficiencies and focus on the effect of mutation instead of the raw editing efficiency. As for the poor prediction performance of more than 3 or 4 mismatches, we think there exists two main factors. Firstly, there is the issue of data size, as the combination of more than 3 mismatches is quite huge for the whole 20 positions, we can only randomly cover a small portion of these combinations. Secondly, take 4-6 mismatches for example, the

average editing efficiencies is very low and centralized (Revision Figure 1), which makes it difficult for the model to distinguish them.

Revision Figure 1. The editing efficiency and off: on-target ratio distribution of 4-6 mismatches.

3) The guides shown in Figure 1b both have two target nucleobases in the "base editing window". This seems to have been designed on purpose and should be explained.

Respond:

Thank you for your comments. We showed new targets as examples in the revised manuscript to avoid misunderstanding.

4) The online tool for prediction of off-target editing is very simple and nice to use. It would perhaps benefit from including off-target search without having to visit the Cas-OFF or Caspritz websites.

Respond:

Thank you for your suggestion. Incorporating off-target search tools into our prediction model would be ideal. However, due to the significant amount of computing resources required, integrating these tools will result in slower response times when using the prediction service on our current website. Therefore, we recommend users obtain off-target sequences from Cas-OFF or Caspritz tools before inputting them into our online model.

Figure 1b The reason for selecting the edits represented in Fig. 1b is not clear and editing efficiency is not given. Please clarify and indicate the corresponding editing efficiencies.

Respond:

Figure 1b is used to illustrate the off-target design in our study. We sampled two of the sgRNAs and aligned the off-targets and on-targets. The corresponding editing efficiencies were shown in the revised manuscript.

Figure 1e and 1f The finding that a 1bp mismatch or a 1 bp insertion at position 20 has almost no impact on base editing activity is totally unexpected. The authors should check the impact of these specific mutations on Cas9 cleavage activity for some of the sequences tested.

Respond:

Thank you for your comments. These data also confused us. We checked the literature, where authors showed that one single mismatch at position 20 did not influence conversion efficiency and indel efficiency (HBB site), while one single mismatch at position 20 strongly influenced conversion efficiency and indel efficiency (EMX1 site)⁷. It seems that single mismatches at positions 12-18 have a stronger influence than those at position 20 (Revision Figure 2)⁷, consistent with our results.

Figure 1 Tolerance of BE3 and Cas9 for mismatched sgRNAs. (a,b) Mismatched sgRNAs that differed from the *EMX1* (a) and *HBB* (b) target sites by 1–4 nucleotides were tested in HEK293T cells. Indel frequencies and cytosine conversion frequencies were measured using targeted deep sequencing. The PAM is shown in blue. Red or black asterisks indicate mismatched sgRNAs that were highly active with BE3 but poorly active with Cas9 or vice versa, respectively. Error bars indicate s.e.m. ($n = 3$).

We checked another literature ⁸, where authors showed that single-mismatches did not influence ABE and CBE efficiency. It seems that the editing reached a plateau. Single mismatches at the seed region influenced indel efficiency at this site (Revision Figure 3).

Fig. 1 | Tolerance of ABE7.10, BE3, and Cas9 for mismatched sgRNAs. Mismatched sgRNAs that differed from the HEK293T site by one to four nucleotides were tested in HEK293T cells. Base editing or indel frequencies were measured using targeted deep sequencing. Mismatched nucleotides and the PAM sequence are shown in red and blue, respectively. Asterisks indicate mismatched sgRNAs whose relative activity (mismatched sgRNA/matched sgRNA) with one enzyme is more than three times higher than that with the other two enzymes. Means \pm s.e.m. were from three independent experiments.

Revision Figure 3, Figure from PMID: 30833658

For another dataset generated in this paper, one single mismatch at position 20 did not influence conversion efficiency and indel efficiency (HEK3 site). Single mismatches at positions 14-16 had a stronger influence than those at positions 18-20. One mismatch at position 20 strongly influenced conversion and indel efficiency (RNF2 site, Revision Figure 4). These data demonstrate that whether single mismatches at position 20 influence editing efficiency depends on the target sequence.

Supplementary Figure 1

Mismatch tolerance of ABE7.10-induced base editing, Cas9-induced indel formation, and BE3-induced base editing in human cells.

sgRNAs targeting the *HEK3* (a) or *RNF2* (b) sites have 0 to 4 mismatches compared with the target sequence. The base editing and indel frequencies were measured using targeted deep sequencing. The red and blue characters depict the mismatched nucleotides and the PAM sequences, respectively. Asterisks indicate mismatched sgRNAs whose relative activity (mismatched sgRNA / matched sgRNA) with one enzyme is more than three times higher than that with the other two enzymes. Means \pm s.e.m. were from three independent experiments.

Revision Figure 4, Figure from PMID: 30833658

We selected seven off-targets with comparable editing efficiency to on-targets from our library and tested their editing efficiency at endogenous loci. Overall, single mismatches at position 20 did not significantly influence off-target efficiency (Revision Figure 5).

a

b

Revision Figure 5. Single mismatches at position 20 did not significantly influence off-target efficiency.

We further investigated whether single mismatches at position 20 influence indel efficiency. Interestingly, single mismatches at position 20 significantly decreased editing efficiency (Revision Figure 6).

Revision Figure 6. Single mismatches at position 20 significantly decreased editing efficiency.

Page 14 Line 280 The method for quantifying conversion efficiencies from Sanger sequencing files should be detailed.

Respond:

Thank you for your comment. I think it was a mistake to put the Sanger sequencing method here. We deleted this part from the revised manuscript.

Page 6 Line 121 Give the exact definition of the seed region used for the analysis.

Respond:

Thank you for pointing this out. The seed region is where 1~9 nucleotides are proximal to the PAM³⁻⁶. We added this explanation in the last paragraph of the first section of the results.

Page 2 Line 50 Typo correction needed. Although there exists (exists) several tools for base editor on-target efficiency.

Respond:

Thank you for your suggestion. We have corrected the spelling errors in the manuscript.

REPLY TO REFEREE #3:

In this work, the authors report on their study of designing gRNA mismatched target libraries and consequently training machine learning models (neural network based model) to predict editing efficiency at off-targets for both (1) Adenine base editor (ABEs) and (2) Cytosine base editors (CBE). The authors reported on the performance of their trained models on held-out test sets (test sets from the 10 times shuffle split of the data) and on assembled data from the literature (i.e. external data). Overall the reported models

achieved good performance on the generated experimental data (i.e. in-house data) and modest to low performance on external data. The authors provide access to the models through public code repository and a web app that accepts a gRNA-off-target pair and outputs the predicted editing efficiency.

While reading the manuscript and inspecting the code, I had couple of comments and suggestions:

1- one main issue in the setup is the data splitting. The authors considers the whole set of sequence pairs (on- and off- target) as the unique "unit of analysis" and then proceed to do splitting of data into training and testing. The problem is that the data is based on pairs from the gRNA-on-target pair (i.e. initial sequences) and then multiple gRNA-off-target pairs associated with them. For example, in case of ABE data, you have 1110 unique sequences (on-target sequences) and 54663 off-target sequences associated with these initial sequences. In a trivial way, when splitting the data we might end up having pairs of same initial sequence ending up in both training and testing. This design would lead to a "leak" as both sequence pairs are correlated — this is amplified given that major off-targets are based on 1bp mismatch, insertion or deletion. To test this hypothesis, if we group the data by these 1110 sequences (i.e. the "source" column in the ABE data frame), and then observe the average editing efficiency within each group, the multiple off-targets are generally close to that mean.

Hence, a more unbiased option is to (1) first group the sequences by the "on-target" sequences (in case of ABE, it would be the 1110 unique sequences in the "source" column) and then perform the data splitting based on these sequences (i.e. the group

number associated to these sequences). In other words, we would have 1110 ids to distribute between training or testing and then all off-target pairs associated with these ids could be used safely as there is no chance of having same initial (on-target) associated to train and test sets.

1 A related comment to the splitting setup is to use 5-fold or 10-fold cross-validation rather than "shuffle split". As mentioned in point 1, after the grouping and doing the splitting based on the unique "source" (on-target) sequences, I suggest to do 5-fold or 10-fold splitting. The reason is that with this setup, all sequence pairs will be in the test set at one point in the evaluation. Whereas the shuffle split does not guarantee that and also does not guarantee that all folds will be different (please see here https://scikit-learn.org/stable/modules/generated/sklearn.model_selection.ShuffleSplit.html). To conclude this point, my suggestion is to use some variation of GroupKFold (https://scikit-learn.org/stable/modules/generated/sklearn.model_selection.GroupKFold.html) where the grouping is based on the unique on-target sequences (i.e. 1110 in the case of ABE).

1.2 Another comment related to the reporting in the writeup about the splitting procedure. The authors refer to the shuffle split as "cross-validation" see page 6 line 131 ".. for tenfold cross-validation" and similarly page 8 line 147. If the authors would continue with the shuffle split approach, I think rewording these sentences would be more accurate.

Respond:

We appreciate the reviewer's instructive suggestion to improve our manuscript. Following the reviewer's suggestion, we grouped the datasets according to on-target sequence and obtained 1,100 ABE gRNA groups and 1,076 CBE gRNA groups, respectively. These

groups were randomly divided into training and testing datasets, with 85% of the gRNA groups assigned to the training set and 15% to the testing set. For ABE, this resulted in 47,974 training gRNA-off-target pairs and 7,799 testing on-off-target pairs, while for CBE, there were 49,371 training on-off-target pairs and 7,432 testing gRNA-off-target pairs. The internal testing datasets were kept fixed and were never used during the training process. Furthermore, we performed 10-fold "GroupKFold" cross-validation on the remaining 85% training gRNA groups instead of using "shuffled split," which could lead to overfitting.

Besides, we should mention that based on the Reviewer 1s' comments, it's more appropriate to use the ratio of off-target efficiency to on-target efficiency (off:on-target ratio) in our study than to use editing efficiency directly, which would have reduced the impact of a wide distribution of on-target edit efficiency on off-target assessment.

2. The authors reported on the design of neural network based model that uses shared embedding matrix and bidirectional RNN layer followed by output regression layer. The design space for the neural network models (i.e. choice of architecture) is huge and there are many variations of computational layers that could be used for modelling the current problem. I think it is unfair to ask for different models (i.e. designs) based on other architectures etc. But I think it is helpful to assess to what degree the neural model (current architecture) is successful in solving the prediction problem relative to a baseline approach. And this could be done comparing it to a baseline to understand the difficulty of the prediction task. For example fitting a gradient boosting based model (such as Xgboost) would be a great baseline to compare the added value of the current neural network architecture.

Respond:

Thank you for your comment. This is the same suggestion as Reviewer 2. Figure 2b displays the schematic of the dataset and algorithms used for benchmarking the off-target predicting task. We compared four conventional hand-crafted feature algorithms: Linear Regression, Ridge Regression, Multiple Perceptron, and XGBoost with the proposed deep learning methods ABEdeepoff and CBEdeepoff, which are primarily constructed using LSTM. Due to the superior feature extraction and representation capabilities as well as the model complexity, the LSTM model for deep learning significantly outperforms the other four traditional algorithms (Supplementary Tables 5 and 7).

3. Looking at Figure 1 C and D, it can be concluded that On, 1mis, 1ins, 1del cases have wider variation of editing efficiency compared to other off-target mutations that cluster at the low editing rate. It would be interesting to inspect the models predictions and highlight what learned features (i.e. position and nucleotides) for a given pair that contribute to the model's decision of increased or decreased editing efficiency prediction. Currently we have descriptive statistics (i.e. distribution) on the different editing rates but not much on the models' prediction process.

Respond:

Thank you for your advice. To understand the feature contribution of the models, we implement the "LayerIntegratedGradient" class from the Captum package⁹ to the embedding layers to evaluate the attribution score for each nucleotide position in the input sequences. The attribution score for each nucleotide position was calculated by taking the mean value of that position across the entire testing dataset. We found that features

at all off-target positions contribute negatively to the final output (ABE, -15.96 ± 7.92 ; CBE, -9.65 ± 4.65), although there are some on-target nucleotide positions that have a negative contribution (Figure 4a-b).

Furthermore, to estimate the contribution of the off-target nucleotides, we comprehensively analyzed the attribution score for 1mis, 1del, and 1ins positions for ABE and CBE, separately. Since the theoretical value of off:on-target ratios are 1 for all the gRNA-off-target pairs, we do not evaluate model feature attribution for on-target pairs data in this study. The attribution scores were standardized by the Z-score, where values higher or lower than 1 signified a significantly greater or lesser contribution than the average, respectively (Figure 4c-d). Overall, off-targeting before the 10th nucleotide has a relatively small impact on the final editing efficiency (z-score above 0 or even 1), but off-targeting after the 10th nucleotide usually has a large negative impact on the editing efficiency. However, for 1ins both in ABE and CBE, we found that the off-target sequences were more tolerant if the insertion position was at 20 or 21. This is consistent with the conclusion we reached in the first section of the result, but the significance degree differs (Supplementary Figure 3). We have added a section named “The interpretability of the models” in Results part to discuss this.

4. When reporting the models performance on experimental test sets, we see average performance of the 10 folds trained models with the standard deviation (Figure 3 A and C). But when it comes to external data, it is not clear if the performance is based on (a) one specific chosen model, or (b) it is the average of predictions from the 10 models (such

as mean prediction) correlated with the ground-truth, or (c) average performance across the 10 fold models. Maybe captioning the figure with this info (or mentioning it in the main manuscript) is helpful for reproducibility.

Respond:

Thank you for your advice. For the external test data, the average predictions from the 10 models were obtained and used to calculate the Spearman correlation score with the ground truth. And we have added a more precise description in the figure legend of Figure 3.

5. Some inconsistencies noted between the manuscript and the code repo shared.

5.1 I noted that during training the batches are not shuffled with every iteration see <https://github.com/izhangcd/BEdeep/blob/767082f0e05488b228783581eb900b3c7a16e5fb/training.py#L350> I think this might be a bug as you typically would like to shuffle the mini batches during training

5.2 For the embedding of sequence pair (i.e. gRNA and off-target), equation 1 (line 341), the authors refer to using the "sum" of both embedding at each position. It might be worth mentioning that `nn.EmbeddingBag` (<https://pytorch.org/docs/stable/generated/torch.nn.EmbeddingBag.html>) used in the code <https://github.com/izhangcd/BEdeep/blob/767082f0e05488b228783581eb900b3c7a16e5fb/training.py#L114> by default uses "mode=mean" (that is it averages rather than summing unless explicitly specified).

5.3 I was wondering about the weights used while computing the loss function. They are specific to each "off type" in the data <https://github.com/izhangcd/BEdeep/blob/767082f0e05488b228783581eb900b3c7a16e5fb/training.py#L24> Generally, you would estimate these from the training data (i.e. training partition) which I assume using inverse ratio of prevalence of each category. But it seems they are given and most probably estimated using the whole data. I do not think this will affect the results or the dynamics of the training, but I bring the authors' attention to this point.

Respond:

Thank you for pointing out these problems. We have revised our code to keep it consistent with the manuscript description. It should also be noted that to perform the feature attribution analysis. We rewrite the shared embedding structure to shared initialized weight (separate embeddings for on-target and off-target but with the same initialized weight setting). And as suggested by the reviewer, to be able to dynamically adjust the loss weight of mutation type in the training data, we use the inverse ratio of prevalence of each category in each batch instead of in the whole data.

6. General comments on the figures:

6.1 For Figure 1 C and D, it might be helpful to add the number of samples on the margin/side (i.e., $n=xx$) to basically complete the picture of how many samples for each distribution - given that there is a huge difference between the categories (i.e., >33000 1mis and the second next category is >6000 for 2mis, etc.)

Respond:

The number of samples has been added to the margins in Figures 1C and 1D.

6.2 The model architecture schematic (Figure 2 A) could become clearer if either the input is (1) one sequence pair or (2) batch sequence pairs (as it is now) but the shared embedding would have a stacked representation (i.e. overlaid) of the different sequence pairs. I say this because it was confusing to follow as it seems the shared embedding part refers to the embedding of each position in one sequence pair rather a batch of sequence pairs — I might be wrong interpreting it so I apologise in advance.

Response:

Thanks for pointing this out. We have recreated Figure 2A to represent a batch of embeddings with stacked representation.

6.3 I was wondering in Figure 1 e and f, why does the editing efficiency bounces back (it increases) when it is at position 20? I know it is near the PAM, but looking at the previously cited paper (ref [19]), it shows that nucleotide at position 20 (typically G) leads to higher editing for a sequence. Does that mean, the 1mis would be having a letter switching to G in these cases, or when it is 1ins it is a G inserted at that position?

Response:

Thank you for your comments. Inspired by your question, we investigated the positional effects of every single-nucleotide mutation on the off:on-target ratio. Naturally, for deletion and insertion mutation, there is only one case, that is, what nucleotide is missing or

inserted. But for mismatch, there are two cases. One analysis perspective is an original nucleotide, and the other analysis perspective is a mutational nucleotide. Here, we mainly focus on the first perspective in mismatch-type mutation.

Similar to the response of 3rd comment, the off:on-target ratio was standardized by the Z-score. The results revealed that ABE and CBE had similar z-score distributions (Supplementary Figure. 3). For a given mutation type, all four nucleotides showed similar z-score distributions. One nucleotide insertion at position 20 had a small influence on editing efficiency (Supplementary Figure. 3). All four nucleotides displayed similar effects.

We also analyzed the second perspective of mismatch mutation type, namely, the mutational nucleotide perspective (Revision Figure 7). We found that for ABE, a nucleotide mutated to C or G will lead to a more negative impact on the editing efficiency than that mutated to A or T (Revision Figure 7). For CBE, however, all four types of mutated nucleotides have a negligible impact on editing efficiency.

Revision Figure 8. The positional effects of mutational nucleotide perspective. The blue, orange, green, and purple columns indicate that other nucleotides changed to A, C, G, and T, respectively.

1. Wang, D. et al. Optimized CRISPR guide RNA design for two high-fidelity Cas9 variants by deep learning. *Nat Commun* **10**, 4284 (2019).
2. Loshchilov, I. & Hutter, F. Decoupled weight decay regularization. *arXiv preprint arXiv:1711.05101* (2017).
3. Jones, S.K., Jr. et al. Massively parallel kinetic profiling of natural and engineered CRISPR nucleases. *Nat Biotechnol* **39**, 84-93 (2021).
4. Jiang, W., Bikard, D., Cox, D., Zhang, F. & Marraffini, L.A. RNA-guided editing of bacterial genomes using CRISPR-Cas systems. *Nat Biotechnol* **31**, 233-239 (2013).
5. Jinek, M. et al. A programmable dual-RNA-guided DNA endonuclease in adaptive bacterial immunity. *Science* **337**, 816-821 (2012).
6. Hsu, P.D. et al. DNA targeting specificity of RNA-guided Cas9 nucleases. *Nat Biotechnol* **31**, 827-832 (2013).
7. Kim, D. et al. Genome-wide target specificities of CRISPR RNA-guided programmable deaminases. *Nat Biotechnol* **35**, 475-480 (2017).

8. Kim, D., Kim, D.E., Lee, G., Cho, S.I. & Kim, J.S. Genome-wide target specificity of CRISPR RNA-guided adenine base editors. *Nat Biotechnol* **37**, 430-435 (2019).
9. Sundararajan, M., Taly, A. & Yan, Q. in International conference on machine learning 3319-3328 (PMLR, 2017).

REVIEWER COMMENTS

Reviewer #1 (Remarks to the Author):

The authors have addressed all my concerns, except for three minor points:

1. Line 227, the author stated that “Spearman correlation scores varying from 0.754 to 0.869”. How was the “0.754” calculated, as it is not in figure 3d? I assume that might be a typo.

2. Line 491, the authors included free energy as a feature for the baseline models but did not specify how the free energy was calculated.

3. For minor comment 2, corrections to two of the references were requested. However, the references in the revised manuscript are the same as the previous ones. Here are two example references for the application of base editor in prokaryotes and plants:

1) Zong Y, Wang Y, Li C, et al. Precise base editing in rice, wheat and maize with a Cas9-cytidine deaminase fusion. *Nat Biotechnol* 2017;35:438–440. DOI: 10.1038/nbt.3811.

2) Banno S, Nishida K, Arazoe T, et al. Deaminase-mediated multiplex genome editing in *Escherichia coli*. *Nat Microbiol* 2018;3:423–429. DOI: 10.1038/s41564-017-0102-6.

Reviewer #2 (Remarks to the Author):

The authors have made many significant and useful improvements to the manuscript.

However, I would hope response to one of my comments (comment 3) could still be improved. I will therefore give an extreme example to help make my point.

For the on-target/off-target pair below, the efficiency prediction given for A to G base editor by the BEdeep software is very high (=0.513347) even though there is no target A base in the off-target sequence and the prediction should therefore be 0 !

GCGTGAGGCTCCGGCGGCGCGG GCGTGtGGCTTCCGGCGGCGCGG 0.513347

Something seems wrong doesn't it ?

Possibly due to the design of datasets, the impact of the presence and position of target nucleobases in the editing window (i.e. A for ABE and C for CBE) is not very well taken into account by the prediction score ?

I therefore initially wondered if this could be analyzed and discussed in the manuscript and whether the prediction score could be improved by providing information on the known impact of the position of target nucleobases on editing efficiency.

Reviewer #3 (Remarks to the Author):

Overall the authors have satisfied most of the comments/questions i raised in the first review. Some additional comments after reading the current manuscript:

- Although, it might not have a huge impact, I would have preferred that the authors trained and tested their models using 5-fold or 10-fold cross validation such that each sample would be in a test set at one point. In other words, once the authors grouped the dataset based on the on-targets, instead of splitting randomly 85% training and 15% testing (fixed set), they could have split using GroupedKfold where you create 10 or 5 folds, each has a (training, testing) set. Then within the training, you can sample a validation set that you can use to tune hyperparameters of each model. I think this would generate less biased estimates of the performance. Maybe such experiment could be added to the supplementary where authors report the performance using this setup (this saves them from changing all figures and writing in the main manuscript).

- Also the performance of the baseline models is striking as there is large difference between the best baseline model and the proposed neural model (in some cases it is more than 60% performance increase !). I wonder if the authors setup a wide array (enough varied values of hyperparameter options for baseline models) to try and test on the validation set? Judging by the performance and the SD bars, it seems there is much variation in performance among baseline models as opposed to the proposed neural model. This suggests that the proposed neural model have the hyperparameters fixed (i.e. for the 10 models trained) and the only variation was the training/validation split they had access to? Where the baseline models, probably there was different hyperparameter choices for each of the folds?

- I would rephrase the use of “significant” when switching to Z scores and considering values above 1 or -1 SD. Traditionally, in a normal distribution, values roughly above 1.96 or -1.96 would correspond to the 5% area of the curve where “significance” is claimed. I would simply define the threshold and call it “important contribution” or similar phrasing.

- I did not follow the argument the authors raised for switching the outcome value from off-target editing to off-target:on-target ratio. The authors mention about the wide range of editing for on-target but that is also the case of off-target editing especially if you look at 1bp misplacement. Also when switching to this setup, are the on-target editing sequences part of the data prediction? Maybe I misunderstood this part.

- I would suggest rephrasing the “interpretability of the models” section into “Model postdoc explainability”

- Minor correction, Line 170 from the “LSMT output”, to be to be corrected to be “LSTM output”

NCOMMS-22-37542B (Prediction of base editor off-target activity by deep learning)

We would like to express our sincere gratitude to the reviewers for their invaluable time and effort in the review process. We greatly appreciate the constructive feedback and helpful suggestions that have significantly improved the quality of our manuscript. A copy of the reviewers' comments, highlighted in grey, is provided below. The comments are followed by a detailed point-by-point response. All changes and additions to the manuscript are indicated for ease of review.

REPLY TO REFEREE #1:

1. Line 227, the author stated that “Spearman correlation scores varying from 0.754 to 0.869”. How was the “0.754” calculated, as it is not in figure 3d? I assume that might be a typo.

Respond:

Thank you for pointing out the typo. According to reviewer 3's comments, we retrained the model using 10-fold cross-validation. Now the Spearman correlation in the external test data set varied from 0.710 (Fig. 3d) to 0.859 (Fig. 3d).

2. Line 491, the authors included free energy as a feature for the baseline

models but did not specify how the free energy was calculated.

Respond:

Thank you for your comments. The free energy, which is considered a more accurate measure of RNA/DNA binding stability, was calculated using the nearest neighbor method ¹. The sgRNA design tool WU-CRISPR also adopts this as one of the features to predict genome-wide sgRNA activity ². We have added a description of how to calculate the binding free energy in the “Comparison with baseline models” subsection of the Methods section.

1. Xia, Tianbing, et al. "Thermodynamic parameters for an expanded nearest-neighbor model for formation of RNA duplexes with Watson–Crick base pairs." *Biochemistry* 37.42 (1998): 14719-14735.

2. Wong, Nathan, Weijun Liu, and Xiaowei Wang. "WU-CRISPR: characteristics of functional guide RNAs for the CRISPR/Cas9 system." *Genome biology* 16.1 (2015): 1-8.

3. For minor comment 2, corrections to two of the references were requested.

However, the references in the revised manuscript are the same as the previous ones. Here are two example references for the application of base editor in prokaryotes and plants:

1) Zong Y, Wang Y, Li C, et al. Precise base editing in rice, wheat and maize with a Cas9-cytidine deaminase fusion. *Nat Biotechnol* 2017;35:438–440. DOI: 10.1038/nbt.3811.

2) Banno S, Nishida K, Arazoe T, et al. Deaminase-mediated multiplex genome

editing in Escherichia coli. Nat Microbiol 2018;3:423–429. DOI:
10.1038/s41564-017-0102-6.

Respond:

Thank you for your suggestion. The two references for the application of base editor have been added to the manuscript.

REPLY TO REFEREE #2:

The authors have made many significant and useful improvements to the manuscript.

However, I would hope response to one of my comments (comment 3) could still be improved. I will therefore give an extreme example to help make my point.

For the on-target/off-target pair below, the efficiency prediction given for A to G base editor by the BEdeep software is very high (=0.513347) even though there is no target A base in the off-target sequence and the prediction should therefore be 0!

GCGTGAGGCTTCCGGCGGCGCGG GCGTGtGGCTTCCGGCGGCGCGG
0.513347

Something seems wrong, doesn't it?

Possibly due to the design of datasets, the impact of the presence and position of target nucleobases in the editing window (i.e., A for ABE and C for CBE) is not very well taken into account by the prediction score?

I therefore initially wondered if this could be analyzed and discussed in the manuscript and whether the prediction score could be improved by providing information on the known impact of the position of target nucleobases on editing efficiency.

Respond:

Thank you for pointing that out. Our library did not include sequences without editable nucleobases. In the revised manuscript, the off:on-target ratio is set to 0 for such sequences. In addition, reviewer 3 also asked about the predicted results of gRNA-on-target pairs, in which the theoretical off:on-target ratio is 1. Thus, we pre-checked the model inputs and enhanced the predicted output of the model in the web server for both two situations (Reviewer Only Figure 1):

1. For sequences without editable nucleobases, the off:on-target ratio is set to 0. The sequence is highlighted in bold and grey in the webserver.
2. For sequences identical to the on-target sequence, the off:on-target ratio is set to 1. The sequence is highlighted in bold and blue in the webserver.

Reviewer Only Figure 1. The non-target and on-target prediction results are on the BEdeepoff website.

We have added a description and discussion of the two situations in the last paragraph of the Results section.

REPLY TO REFEREE #3:

Overall, the authors have satisfied most of the comments/questions I raised in the first review. Some additional comments after reading the current manuscript:

- Although, it might not have a huge impact, I would have preferred that the authors trained and tested their models using 5-fold or 10-fold cross validation such that each sample would be in a test set at one point. In other words, once the authors grouped the dataset based on the on-targets, instead of splitting randomly 85% training and 15% testing (fixed set), they could have split using GroupedKfold where you create 10 or 5 folds, each has a (training, testing) set. Then within the training, you can sample a validation set that you can use to

tune hyperparameters of each model. I think this would generate less biased estimates of the performance. Maybe such experiment could be added to the supplementary where authors report the performance using this setup (this saves them from changing all figures and writing in the main manuscript).

Respond:

Thank you for your suggestion. In this revision, instead of using a fixed testing set, we employed the 10-fold GroupKFold method to split the entire dataset into training and testing sets. As suggested by the reviewer, the internal validation set was randomly sampled from the remaining testing-excluded dataset to tune hyperparameters. All figures pertaining to model training and performance in the manuscript and supplementary files have been updated, except for the “Model Post-hoc Explainability” section (“explanatory model” was randomly chosen from the best models of the 10-fold).

Supplementary Figure 5. 10-fold cross validation in model training and testing.

- Also, the performance of the baseline models is striking as there is large difference between the best baseline model and the proposed neural model (in

some cases it is more than 60% performance increase!). I wonder if the authors setup a wide array (enough varied values of hyperparameter options for baseline models) to try and test on the validation set? Judging by the performance and the SD bars, it seems there is much variation in performance among baseline models as opposed to the proposed neural model. This suggests that the proposed neural model have the hyperparameters fixed (i.e., for the 10 models trained) and the only variation was the training/validation split they had access to? Where the baseline models, probably there was different hyperparameter choices for each of the folds?

Respond:

Thank you for your suggestions. The optimal hyperparameters remain fixed in the 10-GroupKFold cross-validation in baseline and LSTM models. As suggested by the reviewer, we expand the range of hyperparameters for the baseline modes. The detailed hyperparameters were shown in the table below. We use a Bayesian optimization method named TPE (Tree-structured Parzen Estimator) provided by Optuna ¹ package to obtain the optimal hyperparameters. The standard deviations of the baseline models were still higher than that of the proposed neural model.

Supplementary Table 14. The hyperparameters used for Tree-structured Parzen Estimator.

Baseline model	Hyperparameter	Search space
XGBoost Regression	n_estimators	Start=100, End=200, Step=5
	max_depth	Start=10, End=20, Step=1
	max_leaves	Start=10, End=20, Step=1
	learning_rate	Start=0.005, End=0.05, Step=0.005
MLP Regression	hidden_layer_sizes	Start=100, End=400, Step=100
	activation	identity, logistic, tanh, relu
	solver	lbfgs, sgd, adam
	alpha	Start=0.005, End=0.05, Step=0.005
	learning_rate	constant, invscaling, adaptive

1. Akiba, T., Sano, S., Yanase, T., Ohta, T. & Koyama, M. in Proceedings of the 25th ACM SIGKDD international conference on knowledge discovery & data mining 2623-2631 (2019).

- I would rephrase the use of “significant” when switching to Z scores and considering values above 1 or -1 SD. Traditionally, in a normal distribution,

values roughly above 1.96 or -1.96 would correspond to the 5% area of the curve where “significance” is claimed. I would simply define the threshold and call it “important contribution” or similar phrasing.

Respond:

Thank you for your suggestion. We changed the word "significance" to "important contribution".

- I did not follow the argument the authors raised for switching the outcome value from off-target editing to off-target:on-target ratio. The authors mention about the wide range of editing for on-target but that is also the case of off-target editing especially if you look at 1bp misplacement. Also, when switching to this setup, are the on-target editing sequences part of the data prediction? Maybe I misunderstood this part.

Respond:

Thank you for your comment. We apologize for the misunderstanding caused by our statement. Previous studies used the ratio of off-target efficiency to on-target efficiency to evaluate Cas9 specificity ^{1,2}. In line with previous studies, we used the ratio of off-target efficiency to on-target efficiency (off:on-target ratio) to evaluate base editor specificity. This metric made it possible to compare the specificity of all single-guide RNAs. The two gRNA-off-target pairs shown in the Reviewer Only Table 1 were taken from Supplementary Table 1. Although

the off-target editing efficiency is close, the second off:on-target ratio is much higher than the first one due to the difference in their on-target efficiencies. The off:on-target ratio can be used as a better indicator for selecting highly specific gRNAs.

Reviewer Only Table 1. Off-target efficiency and the off:on-target ratio of two gRNA off-target pairs

gRNA Sequence	Target Sequence	Mutation Type	Off-target efficiency	On-target efficiency	Off:on-target ratio
AGTACTCCACGTT GCGTACGA	AGTA ^a TCCACGTT GCGTACGA	1mis	0.130154	0.952 421	0.136656
AGAGGACTCATCT GCTGCTTG	AGAGGAC ^a CATCT GCTGCTTG	1mis	0.144279	0.144 812	0.996319

After switching, on-target sequences are still included in the training. The Reviewer Only Figure 2 shows that the on-target prediction:on-target test ratios are very close to 1 (0.999 ± 0.002 for ABE and 0.999 ± 0.003 for CBE). Therefore, the off:on-target ratio of the gRNA-on-target pair is set to 1 and highlighted in bold and blue on the website (Reviewer Only Figure 3).

Reviewer Only Figure 2. The predicted results of gRNA-on-target pairs for ABE and CBE.

Reviewer Only Figure 3. The on-target prediction result in the BEdeepoff website.

1. Tycko, Josh, et al. "Pairwise library screen systematically interrogates Staphylococcus aureus Cas9 specificity in human cells." *Nature communications* 9.1 (2018): 2962.
2. Jones, Stephen K., et al. "Massively parallel kinetic profiling of natural and engineered CRISPR nucleases." *Biophysical Journal* 120.3 (2021): 138a.

- I would suggest rephrasing the “interpretability of the models” section into “Model postdoc explain ability.”

Respond:

Thank you for your suggestion. We have rephrased the “interpretability of the models” section into “Model postdoc explain ability.”

- Minor correction, Line 170 from the “LSMT output”, to be to be corrected to be “LSTM output.”

Thank you for your suggestion. We have corrected the typo error with the LSTM output.

REVIEWERS' COMMENTS

Reviewer #1 (Remarks to the Author):

The authors have addressed all my concerns.

Reviewer #2 (Remarks to the Author):

The authors have addressed all my concerns. The work reported in the manuscript is an important contribution to the prediction of base editor off-targets.

Reviewer #3 (Remarks to the Author):

The authors have answered most of the comments/questions I raised and the manuscript has improved. However, I might have misunderstood the target outcome prediction this work is optimising. I have raised the question about the ratio off:on target and from reading reviewer 2 comment I have some additional comments/questions that I would like to raise (and I apologise in advance because I might be simply misunderstanding the setup here):

Revisiting the experiment design, we have gRNA-target pairs for ABE base editor and the target base is A->G or T->C transition. Similarly, for CBE base editor, our target base is C->T or G->A transitions. Let's take the case when target transition is A->G using ABE editor, then we generate gRNA-target pair where we target letter A at different position/s and compute the editing efficiency. These are denoted by the "on-target" efficiency (i.e. sequences where the change only occurs when A->G transition occur). Additionally, we pair gRNA-target sequences where we introduce mutations (such as replacement, deletion, insertion) at different positions to the on-target sequence and measure the editing efficiency (this is denoted by off-target efficiency). This is represented in Figure 1 a in the manuscript.

Abstracting the setup, we basically have the following patterns:

gRNA Target

N..A.N..N -> N..G.N..N (on-target editing efficiency)

N..A.N..N -> N..G.M..N (off-target where additionally N->M mutation occurs)

N is any character {A, C, T, G} and M is the off-target mutation. We could have variation on this pattern such as multiple A->G transitions and multiple N->M off-target mutations (with different mutation types such as insertion, deletion or replacement). But in essence we have target letter change and off-target change. If I understood properly, the authors create a ratio outcome by dividing the off-target edit rate with the on-target edit rate. In other words, we use the on-target edit sequence as our "anchor" to contextualise the off-target edit (i.e. relative score) and we mainly use the off-target sequences with this computed ratio to train machine learning models. The authors' response mention that they use the on-target sequences during training. Is it included as I wrote above, or is it included as a training sample by itself? (But then what is the score that you use, is the off:on equal to 0?).

Now beyond what I stated above, the issue starts when we get to patterns that were never observed in the data and that the model might have difficulty to extrapolate from (I post main ones):

gRNA Target

(a) N..A.N..N -> N..A.N..N (no change at all i.e. wild-type)

(b) N..A.N..N -> N..{T,C}.N..N (only change of target letter A to {T or C} but not G) this theoretically should not occur and empirically might have very low probability that can be assumed to be zero

(c) N..A.N..N -> N..G.N..N (on-target change A->G with no off-targets) this is the case of considering on-target sequences in the data, but given that the models are trained on off-targets, these sequences served only to compute the ratio outcome (i.e. use on-target edit in the denominator)

(d) N..A.N..N -> N..A.M..N (only off-target mutation N->M and the target letter A -> A stays same)

(e) K..A.K..K -> K..{T,C}.K..K (K can be {T, C, G} and not the target letter A. In this case, we have only one target letter A that changed to {T or C} and not G (i.e. illegal transition) and the target sequence is now without target base). This represents the example raised by reviewer 2.

(f) K..K.K..K -> N..N.N..N (case of not having target base in the input i.e. no A and it does not matter the target at this point)

There are variations on these patterns but the essence to large extent is captured in the (a) to (f) cases. I believe that these issues could be mitigated by checking the input before model inference as the specification is either invalid or was not part of the training and the model cannot extrapolate. For example, returning a ratio equal 1 for cases where there is no target base change and hence no off-targets is semantically incorrect. In other words, "hacking" the output for invalid input is not necessarily a good strategy. I leave this to the authors to think about and comment/touch on in some way in the manuscript.

Following on the authors' reply for motivating the off:on outcome by citing an example from suppl. material (Reviewer only table 1), I tried the examples they posted using the web platform and I attach the results. While I do understand the idea of using the on-target as a way to contextualise the off-target rates, the model output (screenshot attached) show a different conclusion. For both sequences the model outputs a small ratio and does not distinguish between both cases. It might be one of the examples that the model struggle to produce good result but I am interested if this also applies to other examples in the data where off-target is close but the on-target is much different.

NCOMMS-22-37542C (Prediction of base editor off-target activity by deep learning)

We would like to thank the editor and all the reviewers for their valuable feedback, which we have used to improve the quality of our manuscript. A copy of the reviewers' comments, highlighted in grey, is provided below. The comments are followed by a detailed point-by-point response. All changes and additions to the manuscript are indicated for ease of review.

Reply to REFEREE #3:

The authors have answered most of the comments/questions I raised and the manuscript has improved. However, I might have misunderstood the target outcome prediction this work is optimising. I have raised the question about the ratio off:on target and from reading reviewer 2 comment I have some additional comments/questions that I would like to raise (and I apologise in advance because I might be simply misunderstanding the setup here):

Revisiting the experiment design, we have gRNA-target pairs for ABE base editor and the target base is A->G or T->C transition. Similarly, for CBE base editor, our target base is C->T or G->A transitions. Let's take the case when target transition is A->G using ABE editor, then we generate gRNA-target pair where we target letter A at different position/s and compute the editing efficiency. These are denoted by the "on-target" efficiency (i.e. sequences where the change only occurs when A->G transition occur). Additionally, we

pair gRNA-target sequences where we introduce mutations (such as replacement, deletion, insertion) at different positions to the on-target sequence and measure the editing efficiency (this is denoted by off-target efficiency). This is represented in Figure 1 a in the manuscript.

Abstracting the setup, we basically have the following patterns:

gRNA Target

N..A.N..N -> N..G.N..N (on-target editing efficiency)

N..A.N..N -> N..G.M..N (off-target where additionally N->M mutation occurs)

N is any character {A, C, T, G} and M is the off-target mutation. We could have variation on this pattern such as multiple A->G transitions and multiple N->M off-target mutations (with different mutation types such as insertion, deletion or replacement). But in essence we have target letter change and off-target change. If I understood properly, the authors create a ratio outcome by dividing the off-target edit rate with the on-target edit rate. In other words, we use the on-target edit sequence as our "anchor" to contextualise the off-target edit (i.e. relative score) and we mainly use the off-target sequences with this computed ratio to train machine learning models. The authors' response mention that they use the on-target sequences during training. Is it included as I wrote above, or is it included as a training sample by itself? (But then what is the score that you use, is the off: on equal to 0?).

Respond:

Thank you for your comment. The understanding of **Figure 1a** is completely correct, and its primary purpose is to illustrate that off-target designs can still result in effective editing. However, the later part, especially for the model input, if I didn't misunderstand, deviate from the manuscript. Here, we use **Reviewer Only Figure 1** and **Supplementary Fig. 1** to illustrate the point. In our research, the input for the model is blue block instead of yellow block, which means the model accepts designed gRNA and designed off-target sequence (also including designed on-target sequence) as the input data, the label is the ratio of overall off-target editing efficiency over its on-target editing efficiency. Thus, we are not optimizing the target outcome (i.e., detailed editing outcome) efficiency prediction. Meanwhile, the gRNA-on-target design will lead to on-target editing, and the editing efficiency normalized by itself is 1.

Reviewer Only Figure 1. The deep learning model inputs and outputs.

Supplementary Figure 1. The on and off-target sequence design and overall editing efficiency calculation. (a, b) Off-target sequence was designed according to on-target. After base editing (ABE or CBE), we got various specific editing results and overall editing efficiency.

Now beyond what I stated above, the issue starts when we get to patterns that were never observed in the data and that the model might have difficulty to extrapolate from (I post main ones):

gRNA Target

(a) N..A.N..N -> N..A.N..N (no change at all i.e. wild-type)

(b) N..A.N..N -> N..{T,C}.N..N (only change of target letter A to {T or C} but not G) this theoretically should not occur and empirically might have very low probability that can be assumed to be zero

(c) N..A.N..N -> N..G.N..N (on-target change A->G with no off-targets) this is the case of considering on-target sequences in the data, but given that the models are trained on off-targets, these sequences served only to compute the ratio outcome (i.e. use on-target edit in the denominator)

(d) N..A.N..N -> N..A.M..N (only off-target mutation N->M and the target letter A -> A stays same)

(e) K..A.K..K -> K..{T,C}.K..K (K can be {T, C, G} and not the target letter A. In this case, we have only one target letter A that changed to {T or C} and not G (i.e. illegal transition) and the target sequence is now without target base). This represents the example raised by reviewer 2.

(f) K..K.K..K -> N..N.N..N (case of not having target base in the input i.e. no A and it does not matter the target at this point)

There are variations on these patterns but the essence to large extent is captured in the (a) to (f) cases. I believe that these issues could be mitigated by checking the input before model inference as the specification is either invalid or was not part of the training and the model cannot extrapolate. For example, returning a ratio equal 1 for cases where there is no target base change and hence no off-targets is semantically incorrect. In other

words, "hacking" the output for invalid input is not necessarily a good strategy. I leave this to the authors to think about and comment/touch on in some way in the manuscript.

Respond:

As I stated above, our goal is to predict the ratio of a specific gRNA-off-target design of overall off-target editing efficiency normalized by its on-target editing efficiency, its value can be also understood as mutation tolerance. We trained our model with the designed gRNA-on-target pairs, 1-6 mismatches, 1-2 indels, and mix gRNA-off-target pairs. Thus, the various detailed editing patterns here won't be a problem. In fact, what we are more concerned about is the possibility of off-target editing for a specific gRNA, rather than the detailed off-target editing outcomes (and their efficiencies). I hope this explanation addresses your concerns.

Following on the authors' reply for motivating the off: on outcome by citing an example from suppl. material (Reviewer only table 1), I tried the examples they posted using the web platform and I attach the results. While I do understand the idea of using the on-target as a way to contextualise the off-target rates, the model output (screenshot attached) show a different conclusion. For both sequences the model outputs a small ratio and does not distinguish between both cases. It might be one of the examples that the model struggle to produce good result but I am interested if this also applies to other examples in the data where off-target is close but the on-target is much different.

Respond:

Thank you for your comment. There was a mistake in the Reviewer Only Table 1 from the respond of last round. We confounded the the on-target efficiency and off:on-target ratio, and there is an issue with the configuration of the model on the website server. Now, we have fixed the problem. The two pairs have similar off-target efficiency, but the ratio (i.e., mutation tolerance) have large difference due to the on-target efficiency. And the model predicts that.

Reviewer Only Table 1. Off-target efficiency and the off:on-target ratio of two gRNA off-target pairs

gRNA Sequence	Target Sequence	Type	Off-target efficiency	On-target efficiency	Off.on-target ratio	Predicted value
AGAGGACTCATCTGC TGCTTGGG	AGAGGACaCATCTGC TGCTTGGG	1mis	0.144279	0.144812	0.9963 19	0.950039
GGAACACGATCCATC ACCAGAGG	GGAACACGATCCAgC ACCAGAGG	1mis	0.131097	0.881041	0.1487 98	0.184275

Reviewer Only Figures 2 and 3 show the Concordance Correlation Coefficient (CCC) results from the internal testing data. This statistic measures the agreement between ground truth and predictions. The model performed well for off-targets with 1~2bp insertion, deletion, and mismatch, which aligns with the model's Spearman correlation scores.

Reviewer Only Figure 2. The concordance correlation coefficient of the ABE internal testing data.

Reviewer Only Figure 3. The concordance correlation coefficient of the CBE internal testing data.